# Robust Estimation Under Heterogeneous Corruption Rates

**Syomantak Chaudhuri**
University of California, Berkeley

**Jerry Li**
University of Washington

**Thomas A. Courtade**
University of California, Berkeley

## Abstract

We study the problem of robust estimation under heterogeneous corruption rates, where each sample may be independently corrupted with a known but non-identical probability. This setting arises naturally in distributed and federated learning, crowdsourcing, and sensor networks, yet existing robust estimators typically assume uniform or worst-case corruption, ignoring structural heterogeneity. For mean estimation for multivariate bounded distributions and univariate gaussian distributions, we give tight minimax rates for all heterogeneous corruption patterns. For multivariate gaussian mean estimation and linear regression, we establish the minimax rate for squared error up to a factor of $\sqrt{d}$, where $d$ is the dimension. Roughly, our findings suggest that samples beyond a certain corruption threshold may be discarded by the optimal estimators – this threshold is determined by the empirical distribution of the corruption rates given.

## 1 Introduction

In traditional statistics, we typically rely on the assumption that our data is generated in a "nice", i.i.d. manner from some population distribution. Robust statistics can be seen as a relaxation of these assumptions, aiming to ensure meaningful performance even when the data that has been corrupted by a small fraction of arbitrary, but potentially adversarially chosen, outliers. First proposed in seminal work by statisticians such as Huber, Tukey, and Anscombe more than fifty years ago [1–4], this problem remains highly relevant to this day, and has received considerable recent attention from the statistics, machine learning, and theoretical computer science community, see e.g. [5–20]. A full survey of this line of work is beyond the scope of this paper, see e.g. [21] for a more comprehensive overview.

There are a number of related ways in the literature to model such outliers. For instance, one classical setting is known as the *Huber contamination model* [22]. Here, there is a "nice" distribution $D$, typically assumed to be from some well-behaved class of distributions, and we receive $n$ samples $Z_1, \ldots, Z_n$ from a distribution $D' = (1 - \epsilon)D + \epsilon N$ for some arbitrary noise distribution $N$, and some error parameter $\epsilon > 0$. In other words, to generate a sample point $Z_i$, we generate a clean sample from $D$ with probability $(1 - \epsilon)$, and we sample an outlier from a noise distribution with probability $\epsilon$. There are also a number of other, more powerful, adversaries considered in the literature, for instance, which allow the adversary to choose the corrupted points adaptively based on the clean samples; see e.g. [23–25] for a more detailed discussion. However, by now, at least for many basic estimation tasks such as mean estimation, tight (or nearly tight) rates are known for many parametric families of distributions under all of these corruption models.

But, all of these models share an unfortunate drawback: namely, they all essentially assume a homogeneous prior on the error rates across the dataset. In the Huber contamination model, for

example, every sample is corrupted with the same probability $\epsilon$. In practice, this is usually not the case: we often obtain data from various heterogeneous sources, and consequently we should have different priors on the cleanliness of different data points in our dataset. As an example, many large-scale biomedical datasets are agglomerated in a distributed fashion over different institutions and at different points in time, and it is unreasonable to assume that the rate of outliers ought to be similar over the various sub-parts of the data. But because all existing robust statistics techniques assume a uniform noise prior, it is unclear whether can obtain the optimal statistical rates for such datasets, or what these rates even are in the first place.

Motivated by these considerations, we propose a natural generalization of the Huber contamination model with heterogeneous corruption rates, where every data point is corrupted with a potentially different error rate. Formally:

**Definition 1** ($\boldsymbol{\lambda}$-contamination). Let $n$ be the number of samples, and let $\boldsymbol{\lambda} = (\lambda_1, \ldots, \lambda_n) \in [0, 1]^n$. Let $P$ be some distribution. We say a dataset $\boldsymbol{Z} = (Z_1, \ldots, Z_n)$ is a $\boldsymbol{\lambda}$-*contaminated* set of samples from $P$ if $Z_i = (1 - B_i)X_i + B_i\tilde{X}_i$, where:

- $X_1, \ldots, X_n \overset{\text{iid}}{\sim} P$ is a set of clean samples from $P$,
- $B_i \sim \text{Bern}(\lambda_i)$ are independent Bernoulli random variables for $i = 1, \ldots, n$, and
- The outliers $(\tilde{X}_1, \ldots, \tilde{X}_n)$ are sampled from some joint distribution conditioned on the realizations of $X_1, \ldots, X_n$ and $B_1, \ldots, B_n$.

When a set of samples is generated this way, we denote this by $\boldsymbol{Z} \sim_{\boldsymbol{\lambda}} P$.

In other words, we assume that the probability that the $i$-th sample is corrupted is $\lambda_i$, for different values of $\lambda_i$. We pause to make a couple of remarks on this definition. First, when all the $\lambda_i$ are the same, this is essentially the standard Huber contamination model, except we allow that the outliers may be chosen adaptively. Second, we will assume that we know $\boldsymbol{\lambda}$ exactly. However, our upper bounds also naturally generalize to the setting where we only have approximate knowledge of the $\lambda_i$, i.e., we just need a constant factor approximation $\hat{\lambda}_i$ of the corruption rate for our results to hold (i.e., $\lambda_i \leq c\hat{\lambda}_i$). Such upper bounds on the corruption rates can often be estimated from past performance in practice. Finally, just as in standard robust statistics, there are many similar but not equivalent definitions we can consider here, for instance, corresponding to oblivious contamination or strong corruption [21, 23–25]. Our proposed error upper and lower bounds hold for both adaptive and non-adaptive adversaries. We leave the exploration between the differences between these definitions as an interesting future direction.

From a more conceptual point of view, the main challenge in this heterogeneous setting is how to effectively "mix" the information from data points with different noise levels. As an illuminating example, one important special case of the $\boldsymbol{\lambda}$-contamination model is *semi-verified* learning [7, 9], where there is a (very) small amount of trusted data, and a large amount of potentially very noisy data, which corresponds to the setting where $\lambda_i = 0$ for a small number of $i$, and $\lambda_i = \gamma$ for some $\gamma$ close to 1 for the rest of the data. In general, robust estimation is impossible if $\lambda_i \geq 1/2$ for all $i$, however, it turns out that in the semi-verified setting, one can obtain consistent estimators by combining the information from the two sets of samples [7]. This example illustrates the main technical challenge of the general heterogeneous setting. To obtain the tight rates, one must correctly identify how to incorporate information from the noisier samples into the information from cleaner ones, and to do so smoothly as a function of $\lambda_i$.

## 1.1 Our results

We establish lower and upper bounds for the statistical rates for a number of fundamental estimation tasks in the $\boldsymbol{\lambda}$-contaminated setting. We first consider robust mean estimation in the heterogeneous setting, one of the most important and well-studied problems in robust statistics. For any distribution $P$, let $\mu_P = \mathbb{E}_{X \sim P}[X]$ be the expectation of $P$. We will measure the effectiveness of our estimators using the following minimax metrics:

**Definition 2** (Minimax, Minimax PAC rates for heterogeneous robust mean estimation [26, 27]). Let $\mathcal{D}$ be a set of distributions over $\mathbb{R}^d$, and let $\boldsymbol{\lambda} \in [0, 1]^n$. The *minimax rate for $\boldsymbol{\lambda}$-corrupted mean estimation* for the class $\mathcal{D}$ is defined to be

$$L(\boldsymbol{\lambda}, \mathcal{D}) = \inf_M \sup_{P \in \mathcal{D}} \mathbb{E}_{\boldsymbol{Z} \sim_{\boldsymbol{\lambda}} P} \left[ \|M(\boldsymbol{Z}) - \mu_P\|_2^2 \right] , \tag{1}$$

where the infimum is taken over all estimators $M$. The *minimax PAC rate for $\boldsymbol{\lambda}$-corrupted mean estimation* for the class $\mathcal{D}$ is defined to be

$$L_{\mathsf{PAC}}(\boldsymbol{\lambda}, \mathcal{D}, \delta) = \inf_{M} \sup_{P \in \mathcal{D}} \inf \left\{ t \in [0, \infty) : \Pr_{\boldsymbol{Z} \sim_{\boldsymbol{\lambda}} P} \left[ \|M(\boldsymbol{Z}) - \mu_P\|_2^2 \geq t \right] \leq \delta \right\} . \tag{2}$$

Readers can refer to Ma et al. [26] for more details on the minimax PAC rate. Stated simply, the minimax PAC rate is simply the smallest rate so that the probability that the estimator exceeds $L_{\mathsf{PAC}}(\boldsymbol{\lambda}, \mathcal{D}, \delta)$ is small. Since by standard techniques, we can boost the error probability $\delta$, we will typically focus on the setting where $\delta$ is a small constant, and we will let $L_{\mathsf{PAC}}(\boldsymbol{\lambda}, \mathcal{D}) :-$ $L_{\mathsf{PAC}}(\boldsymbol{\lambda}, \mathcal{D}, 1/5)$. We use this notion because it is necessary in the robust setting. In settings where the unknown mean could be unbounded, no estimator can achieve finite expected error, even in the standard Huber contamination model. This is because all samples may be corrupted with exponentially small probability, in which case the error could be arbitrarily large.

**Mean Estimation for Bounded Distributions:** We first consider the class of bounded, multivariate distributions. For this setting, we show:

**Theorem 1.** *Let $\mathcal{D}_r^b$ be the set of all distributions on $\mathbb{R}^d$ supported on the $l_2$ ball of radius $r$. Then,*

$$L(\boldsymbol{\lambda}, \mathcal{D}_r^b) \simeq r^2 f(\boldsymbol{\lambda}, 1) , \text{ where } f(\boldsymbol{\lambda}, k) = \min_{t \in [0,1]} \left( \frac{k}{|\{i : \lambda_i \leq t\}|} + t^2 \right) . \tag{3}$$

*Moreover, the optimal estimator can be implemented in nearly-linear time.*

The function $f(\boldsymbol{\lambda}, k)$ can be thought of as a measure of the "effective" error rate of the dataset, and will play a crucial role in many of our results going forward. Indeed, Theorem 1 implies that the optimal robust mechanism chooses a corruption level $t$ and only utilizes the data having corruption rate below $t$, i.e., there is no improvement by considering the other samples up to constant factors.

**Mean Estimation for Gaussians:** Our next result concerns heterogeneous mean estimation for Gaussians. Our main result can be summarized as follows:

**Theorem 2** (informal, see Theorem 4). *Let $\mathcal{D}_d^{\mathcal{N}}$ the set of all multivariate Gaussian distributions on $\mathbb{R}^d$ with identity covariance. Suppose that $f(\boldsymbol{\lambda}, d) = O(1)$, then $d^{-1/2} f(\boldsymbol{\lambda}, d) \lesssim L_{\mathsf{PAC}}(\boldsymbol{\lambda}, \mathcal{D}_d^{\mathcal{N}}) \lesssim f(\boldsymbol{\lambda}, d)$.*

In other words, we show that, in the regime where there are sufficiently many relatively clean samples, $f(\boldsymbol{\lambda}, d)$ dictates the minimax rate, up to a $\sqrt{d}$ factor. Note that this is arguably the most interesting regime from a statistical perspective, as it is the only regime where the recovered Gaussian has non-trivial statistical overlap with the true Gaussian. We note that our lower bound technique also yields non-trivial bounds in more general settings, however, they are somewhat more difficult to interpret—see the supplementary material for more discussion.

We pause to make a couple of remarks on this result. First, in constant dimensions, our bound is tight up to constant factors, so our bound is tight in this regime. Second, we note that naïve multi-dimensional techniques such as coordinate-wise methods would lose a $\Theta(d)$ factor in the squared $\ell_2$-error, so our bound represents a polynomial improvement over baseline methods.

**Linear Regression:** Finally, we turn our attention to Gaussian design linear regression with heterogeneous corruptions. We assume the uncorrupted covariates are Gaussian with covariance matrix $\Sigma \in \mathbb{R}^{d \times d}$ and noise rate $\sigma^2$. Then, for any choice of regression coefficients $\beta \in \mathbb{R}^d$, we assume that the clean data is of the form $(W, Y) \in \mathbb{R}^{d+1}$ where $W \sim \mathcal{N}(0, \Sigma)$, and conditioned on $W$, $Y \sim \mathcal{N}(W^T \beta, \sigma^2)$. We let $\mathcal{D}(\Sigma, \sigma^2)$ the family of distributions over $(W, Y)$ of this form. For any $P \in \mathcal{D}(\Sigma, \sigma^2)$, we let $\beta_P$ denote its associated regression coefficients. In analogy to Equation (2), we define the minimax PAC rate of squared excess risk under heterogeneous corruptions to be

$$L_{\mathsf{reg}}(\boldsymbol{\lambda}, \mathcal{D}(\Sigma, \sigma^2), \delta) = \inf_{M} \sup_{P \in \mathcal{D}(\Sigma, \sigma^2)} \inf \left\{ t \in [0, \infty) : \Pr_{\boldsymbol{Z} \sim_{\boldsymbol{\lambda}} P} \left[ \|M(\boldsymbol{Z}) - \beta_P\|_{\Sigma}^2 > t \right] \leq \delta \right\} , \tag{4}$$

and as before, we let $L_{\mathsf{reg}}(\boldsymbol{\lambda}, \mathcal{D}(\Sigma, \sigma^2)) :- L_{\mathsf{reg}}(\boldsymbol{\lambda}, \mathcal{D}(\Sigma, \sigma^2), 1/5)$. Here $\|x\|_{\Sigma} = x^\top \Sigma x$, which is known to be the natural affine-invariant measure of error for linear regression [28]. Note that this

is the standard joint contamination model [10, 29], where both the covariate and the target can be jointly corrupted by the adversary, i.e., $Z_i = (W_i(1 - B_i) + B_i\tilde{W}_i, Y_i(1 - B_i) + B_i\tilde{Y}_i)$. This is opposed to the simpler target contamination model also considered in the literature where only the target $Y$ can be corrupted [30].

Our main result in this setting is as follows:

**Theorem 3** (informal, see Theorem 5). *Suppose that* $f(\boldsymbol{\lambda}, d) = O(1)$, *then* $d^{-1/2}f(\boldsymbol{\lambda}, d) \lesssim L_{\mathsf{reg}}(\boldsymbol{\lambda}, \mathcal{D}(\Sigma, \sigma^2)) \lesssim f(\boldsymbol{\lambda}, d)$.

In other words, as before, we establish the tight rate up to a factor of $O(\sqrt{d})$.

**Upper bound techniques**    For all four problems, the upper bounds in the theorem statements can be achieved by a thresholding estimator – set an appropriate threshold $t$ and take the optimal robust mean estimator with homogeneous corruption rate $t$. Thus, at least for bounded distributions and Gaussians in a constant number of dimensions, our results show that there is no significant benefit in collecting data with higher corruption rate beyond a certain threshold.

**Per-sample reweighting**    In addition to this simple thresholding method, we also give a family of more refined estimators that find an optimal per-sample weighting scheme, that match the theoretical guarantees of the simple thresholding estimator, but which has a number of additional advantages over it. First, as we discuss below, these methods seem to sometimes perform better in practice in preliminary synthetic evaluations. Second, in the high-dimensional settings, the thresholding-based methods have error rates which seem to plateau, resulting a the $\sqrt{d}$ gap in the upper and lower bounds. We believe resolving this gap is a very interesting open question. As a first step, we derive natural heterogeneous variants of Tukey depth [4] and regression depth [31] using these per-sample reweighting methods. We believe that these methods may allow us to bridge this gap, by leveraging the higher corruption points that the thresholding-based methods ignore to boost the accuracy. To reiterate, we propose the reweighted estimators as a possible avenue to improve the upper bounds but the estimators used to obtain the minimax upper bounds are based on the thresholding estimators.

**Lower bound techniques**    A key technical challenge that separates our analysis from that of standard homogeneous robust statistics analysis is that of capturing the heterogeneity in the lower bound. The standard approach for the lower bound in homogeneous robust statistics is to add a term due to the corruption rate to the non-robust lower bound [32]. For example, in robust $d$-dimensional Gaussian mean estimation under identity covariance with $\epsilon$ corruption, the minimax rate is of the order $\Theta(\frac{d}{n} + \epsilon^2)$, where $n$ is the number of samples. The term $\frac{d}{n}$ is obtained from classical statistics literature while the $\epsilon^2$ term captures the fact that two distributions with mean within $l_2$ distance of $\epsilon$ can not be distinguished due to the $\epsilon$ contamination – proved via Le Cam's method [28]. Such a two-staged approach is unsuitable for our setting since the number of samples $n$ can be made arbitrarily large by artificially appending samples with $\lambda = 1$. Le Cam's method captures the difficulty of the problem in terms of corruption rate but fails to capture the dimensionality of the problem, while Assouad's method can capture the difficulty of the problem in higher dimension, but seems to have difficulty capturing the power of an adversary. Thus, the challenge lies in constructing tight lower bounds that incorporate both the terms jointly. Readers may refer to our lower bound construction in Appendix C.2 for more details.

**Experimental evaluations**    While we emphasize that the main contribution of this work is theoretical, in the supplementary material, we also perform some preliminary synthetic evaluations to validate the effectiveness of our methods. In the bounded and univariate Gaussian settings, we demonstrate that both the thresholding-based methods as well as the per-sample reweighting methods outperform baselines from the standard homogeneous robust statistics literature. Our results also demonstrate that in some settings, the per-sample reweighting methods also yield improvements over the threshold-based methods in practice.

## 1.2    Related Works

As discussed previously, the robust statistics literature has typically focused on a homogeneous corruption rate. In terms of heterogeneity, Charikar et al. [7] introduced the notion of semi-verified

learning where a small amount of data is sampled from the true distribution in conjunction with a dataset having only $\alpha$ fraction of the data uncorrupted ($\alpha \ll 1$). The semi-verified learning problem is closely related to the problem of list-decodable estimation introduced by Balcan et al. [33]; list-decodable learning aims to recover a list of possible values of the quantity being estimated with the guarantee that one of entries in the list is close to the true value. Typical results in semi-verified learning lead of an error that depends on $\alpha$ but does not typically scale with the number of clean samples. There has been a line of work extending their results [9, 13, 14, 19, 34].

Robustness and privacy are known to be deeply related [35–38]. In the privacy literature there has been a line of work on optimal mechanisms under heterogeneous privacy demands [39–41]. However, these works only focus on bounded univariate setting, and do not seem to imply anything directly for the heterogeneous corruption setting we consider.

*Organization:* We define the problem of mean estimation for bounded distributions in Section 2, and provide a general proof sketch to obtain the minimax rate. In Section 3, we present our results on Gaussian mean estimation for both univariate and multivariate setting, followed by results on excess risk minimization for Gaussian linear regression in Section 4. We present our thoughts on future works in Section 5. Some illustrative experiments can be found in Appendix A.

## 2   Mean Estimation for Bounded Distributions

In this section, we prove Theorem 1.

### 2.1   Upper Bound

To obtain an upper bound on $L(\boldsymbol{\lambda}, \mathcal{D}_r^b)$, we shall restrict our attention to the class of estimators of the form $\sum_{i=1}^n w_i Z_i$ for $w \in \Delta_n$. Via standard bias-variance decomposition,

$$\mathbb{E}\left[\left\|\sum_{i=1}^n w_i Z_i - \mu_P\right\|_2^2\right] = \left\|\sum_{i=1}^n w_i \mathbb{E}[Z_i] - \mu_P\right\|_2^2 + \mathbb{E}\left[\left\|\sum_{i=1}^n w_i(Z_i - \mathbb{E}[Z_i])\right\|_2^2\right]. \tag{5}$$

Denoting $E[\tilde{X}_i | B_i = 1] = \mu_{Q_i}$, the bias term can be upper bounded as

$$\left\|\sum_{i=1}^n w_i(\mathbb{E}[Z_i] - \mu_P)\right\|_2^2 = \left\|\sum_{i=1}^n w_i \lambda_i(\mu_{Q_i} - \mu_P)\right\|_2^2 \leq 4r^2(w^T\lambda)^2, \tag{6}$$

using Jensen's inequality and the fact that $\|\mu_{Q_i} - \mu_P\|_2 \leq 2r$. The variance term needs to be bounded with more care since $\tilde{X}$ are not independent can dependent on $X, B$. It can be upper bounded as $\mathbb{E}\left[\|\sum_{i=1}^n w_i(Z_i - \mathbb{E}[Z_i])\|_2^2\right] \leq 7r^2\|w\|_2^2 + 16r^2(w^T\lambda)^2$ (see Appendix B.1). Thus, we get the upper bound

$$\mathbb{E}\left[\left\|\sum_{i=1}^n w_i Z_i - \mu_P\right\|_2^2\right] \leq 7r^2\left(\|w\|_2^2 + 3(w^T\lambda)^2\right) \;\forall w \in \Delta_n. \tag{7}$$

Taking minimum over $w \in \Delta_n$,

$$L(\boldsymbol{\lambda}, \mathcal{D}_r^b) \leq 7r^2 \min_{w \in \Delta_n} \|w\|_2^2 + 3(w^T\lambda)^2. \tag{8}$$

Since Slater's condition holds, by KKT we obtain the solution of the above to be of form $w_i = (\beta - 3(w^T\lambda)\lambda_i)_+$, where $(x)_+ = \max\{x, 0\}$, for suitable $\beta$ that ensures $w \in \Delta_n$. Exploiting this structure, we obtain a near-linear time algorithm to find the exact minimizer of $\|w\|_2^2 + c(w^T\lambda)^2$ presented in Algorithm 1. For a vector $\lambda$, the notation $\lambda_a^b$ denotes the sub-vector $(\lambda_a, \ldots, \lambda_b)$ for $a \leq b$. The proof of correctness is presented in Appendix B.2.

**Simpler Upper Bound:**   While Algorithm 1 recovers the exact minimizer for (8), we also discuss a significantly simpler upper bound – take the mean of the samples with corruption probability less

---

**Algorithm 1** Robust Mean Estimation for Bounded Distributions

---

Input: $\mathbf{Z}$ and corresponding $\lambda$, with $\lambda$ assumed to be in ascending order
$n \leftarrow \text{LENGTH}(\lambda)$
$k \leftarrow 1$
**while** $k \leq n$ **do**
   **if** $\lambda_{k+1} < \frac{1+c\|\lambda_1^k\|_2^2}{c\|\lambda_1^k\|_1}$ **then**
      $k \leftarrow k+1$
   **else**
      break
   **end if**
**end while**
$\beta \leftarrow \frac{1+c\|\lambda_1^k\|_2^2}{k(1+c\|\lambda_1^k\|_2^2)-c\|\lambda_1^k\|_1^2}, \ \alpha \leftarrow \frac{c\|\lambda_1^k\|_1\beta}{1+c\|\lambda_1^k\|_2^2}.$
$w_1^k \leftarrow \beta - \alpha\lambda_1^k, \ w_{k+1}^n \leftarrow \mathbf{0}$
**return** $\sum_{i=1}^n w_i Z_i$

---

than $t$ for a hyperparameter $t$. Let $N(t) = |\{i : \lambda_i \leq t\}|$. Therefore, for any $t \in [0,1]$, we get the upper bound from (7)

$$L(\lambda, r) \lesssim r^2 \left( \frac{1}{N(t)} + t^2 \right). \tag{9}$$

We can similarly take a minimum over $t$ to get an upper bound on $L(\boldsymbol{\lambda}, \mathcal{D}_r^b)$. This simpler upper bound suffices for proving minimax optimality.

## 2.2 Lower Bound

We use Le Cam's two-point method to obtain the lower bound. Fix some $0 \leq \delta \leq \frac{1}{4}$. Let $P_0 = re_1(2\text{Ber}(\frac{1}{2} - \delta) - 1)$ and $P_1 = re_1(2\text{Ber}(\frac{1}{2} + \delta) - 1)$, where $e_1$ is a unit vector; we have $\|\mu_{P_1} - \mu_{P_2}\|_2^2 = 4r^2\delta^2$.

Under distribution $P_0$, consider the strategy such that adversary tries to ensure that the $i$-th sample is as close to $re_1(2\text{Ber}(\frac{1}{2}) - 1)$ as possible in mean. Let $Q_i = re_1(2\text{Ber}(\gamma_i) - 1)$ with

$$\gamma_i = \begin{cases} \frac{1}{2} - \delta + \frac{\delta}{\lambda_i} & \text{if } \lambda_i \geq \frac{2\delta}{1+2\delta}, \\ \frac{1}{2} - \delta & \text{else,} \end{cases} \tag{10}$$

then the adversary simply samples $\tilde{Y}_i$ independently from $Q_i$. Thus, the adversary can simulate[1] the distribution $re_1(2\text{Ber}(\frac{1}{2}) - 1)$ when $\lambda_i \geq \frac{2\delta}{1+2\delta}$. If $\lambda_i < \frac{2\delta}{1+2\delta}$, then the adversary does not perturb the distribution at all, or equivalently, samples $\tilde{X}_i$ from $P_0$ independently. Thus, the proposed adversarial strategy results in $Z_i$ drawn independently from $re_1(2\text{Ber}(\frac{1}{2} - \epsilon_i) - 1)$, where

$$\epsilon_i := \delta\mathbb{I}\left\{\lambda_i < \frac{2\delta}{1+2\delta}\right\}. \tag{11}$$

Similar arguments hold when the underlying distribution is $P_1$.

Let $n(t) = |\{i : \lambda_i < t\}|$. Using Le Cam's method, we show in Appendix B.3

$$L(\lambda, r) \geq r^2\delta^2 \left(1 - \sqrt{6\delta^2 n(2\delta)}\right) \ \forall \delta \in \left[0, \frac{1}{4}\right] \tag{12}$$

## 2.3 Proving Minimax Optimality

We outline how to pick a 'good' $\delta$. For $x \in [0, \frac{1}{4}]$, let $h(x) = N(2x) - \frac{1}{12x^2}$. Note that $h$ is monotone, piece-wise continuous, and $\exists \alpha > 0$ such that $h(\alpha) < 0$. We use the notation $h(x^-)$ to denote the left limit of $h$ at $x$; note that $h(x^-) = n(2x) - \frac{1}{12x^2}$.

---

[1] By simulate, we mean the mixture distribution matches the claimed distribution.

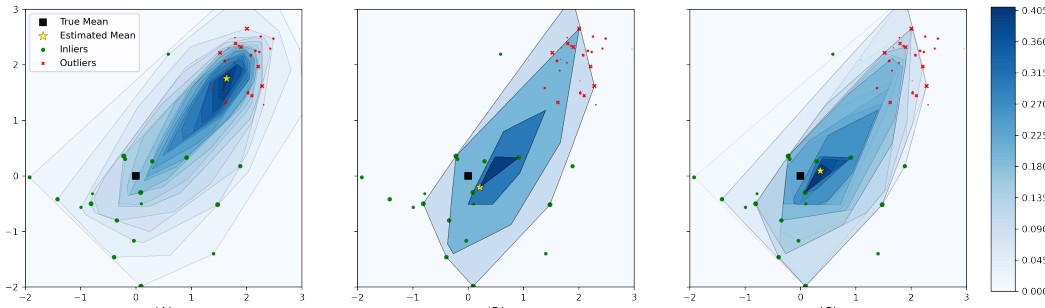

Figure 1: Plot of weighted Tukey depth (see (17)) visualized for three different weighing schemes. (A) is computed with the standard uniform weights $w_i = \frac{1}{n}$, (B) is computed with $w_i = \frac{\mathbb{I}\{\lambda_i \leq t\}}{|\{j:\lambda_j \leq t\}|}$ using the value of $t$ from (21), and (C) is computed with weights given by Algorithm 1. For the dataset, the true underlying distribution is $\mathcal{N}((0,0), I)$, and $\lambda$ is sampled i.i.d. Points are contaminated by replacing them with samples from $\mathcal{N}((2,2), I/5)$. The samples are marked in red 'x' if they were contaminated; the size of the markers for each point is proportional to $1 - \lambda_i$. The estimated mean, the point with maximum depth, is marked with a yellow star.

Consider the case that $h(\frac{1}{4}) < 0$, i.e., $|\{i : \lambda_i \leq \frac{1}{2}\}| < \frac{4}{3}$. This conditions correspond to extremely high levels of corruption and unsurprisingly, we show the minimax rate to be $\Theta(r^2)$. Plug-in $\delta = \frac{1}{4}$ in (12) to get $L(\lambda, r) \gtrsim r^2$. We can get matching upper bound in (9) by setting $t = 1$.

Alternately, if $h(\frac{1}{4}) > 0$ then since $h(\cdot)$ is a *cadlag* function and $h(\alpha) < 0$, there exists $\delta_* \in (0, \frac{1}{4}]$ such that $h(\delta_*) \geq 0$ and $h(\delta_*^-) \leq 0$, i.e., $N(2\delta_*) \geq \frac{1}{12\delta_*^2}$ and $n(2\delta_*) \leq \frac{1}{12\delta_*^2}$. Plug-in $\delta_*$ in (12) to get

$$L(\lambda, r) \geq r^2 \delta_*^2 (1 - \sqrt{6\delta_*^2 n(2\delta_*)}) \tag{13}$$

$$\geq r^2 \delta_*^2 \left(1 - \sqrt{\frac{1}{2}}\right) \simeq r^2 \delta_*^2. \tag{14}$$

In the upper bound of (9), set $t = 2\delta_*$ to get

$$L(\lambda, r) \lesssim r^2 \frac{1}{N(2\delta_*)} + r^2 \delta_*^2 \tag{15}$$

$$\leq 12 r^2 \delta_*^2 + r^2 \delta_*^2 \simeq r^2 \delta_*^2, \tag{16}$$

proving minimax optimality of the proposed schemes in (8) and (9) up to constant factors.

## 3 Mean Estimation for Gaussian Distributions

In this section, we describe our results for the problem of mean estimation for multivariate Gaussian distributions at a high level. For conciseness, we defer the complete proofs to the appendix.

**Upper Bound:** We propose a weighted version of the Tukey median [4]. Define the weighted depth of a point $\eta \in \mathbb{R}^d$ for a dataset $\boldsymbol{Z} = (Z_1, \ldots, Z_n)$ and $w \in \Delta_n$ as

$$D_w(\eta, \boldsymbol{Z}) = \min_{v \in \mathbb{S}_d} \sum_{i=1}^{n} w_i \mathbb{I}\{v^T(Z_i - \eta) \geq 0\}, \tag{17}$$

where $\mathbb{S}_d = \{x \in \mathbb{R}^d : \|x\|_2 = 1\}$. The weighted Tukey median is defined as

$$\hat{\mu}_{\mathsf{TM}}(\boldsymbol{Z}, w) := \arg\max_{\eta \in \mathbb{R}^d} D_w(\eta, \boldsymbol{Z}). \tag{18}$$

The standard Tukey depth (i.e., $w = 1/n$) of a point $\eta$ is the minimum number of samples in a closed half-space with $\eta$ at the boundary. Thus, depth of $\eta$ is high if it is surrounded by samples in all

directions. Our weighted version allows us to decrease the sensitivity to samples which are more likely to be corrupted. In Figure 1, we illustrate the Tukey depth map for three different weighing schemes which demonstrates how using the weights $w$ can be used to leverage the heterogeneity. Proposition 1 establishes an upper bound on the performance of the weighted Tukey median.

**Proposition 1** (Upper Bound for Gaussian Distributions)**.** *For all $w \in \Delta_n$ satisfying $(w^T \lambda)^2 + d\|w\|_2^2 \leq c$ for some universal constants $c$,*

$$\sup_{P \in \mathcal{D}_d^{\mathcal{N}}} \Pr_{\mathbf{Z} \sim_{\lambda} P} \left[ \|\hat{\mu}_{\mathsf{TM}}(\mathbf{Z}, w) - \mu_P\|_2^2 \geq c' \left( (w^T \lambda)^2 + d\|w\|_2^2 \right) \right] \leq 1/5 , \tag{19}$$

*for some universal constant $c'$.*

The proof of Proposition 1 (see Appendix C.1) involves proving that the depth of any point sufficiently far from the true mean is going to be lower than the depth of the true mean in a uniform convergence sense. Due to the weighted nature of the depth definition, we modify results from empirical process theory to adapt to the weights and get tight upper bound.

Let $N(t) = |\{i : \lambda_i \leq t\}|$ and denote the weights $w(t)$ such that $w(t)_i = \frac{\mathbb{I}\{\lambda_i \leq t\}}{N(t)}$, i.e., we consider the Tukey median estimator obtained by only considering samples with corruption less than $t$. Define

$$\hat{\mu}_{\mathsf{S}}(\mathbf{Z}, t) = \hat{\mu}_{\mathsf{TM}}(\mathbf{Z}, w(t)). \tag{20}$$

By Proposition 1, the following holds for some universal constant $c'$

$$\sup_{P \in \mathcal{D}_d^{\mathcal{N}}} \Pr_{\mathbf{Z} \sim_{\lambda} P} \left[ \|\hat{\mu}_{\mathsf{S}}(\mathbf{Z}, t) - \mu_P\|_2^2 \geq c' \left( t^2 + \frac{d}{N(t)} \right) \right] \leq 1/5, \tag{21}$$

$\forall \, t$ such that $t^2 + \frac{d}{N(t)} \leq c$ for universal constants $c$.

## 3.1 Lower Bound and Minimax Rate

**Univariate Case**  We begin our exposition by considering our result for $d = 1$, i.e., univariate Gaussian distributions. We use Le Cam's method to obtain the lower bound [28]; Le Cam's method lower bounds the estimation problem by a hypothesis testing problem where the optimal test is entirely determined by the distribution of data from the different hypotheses. The two hypotheses we consider are $\mathcal{N}(\delta, 1)$ and $\mathcal{N}(-\delta, 1)$ for $\delta$ to be chosen later. The adversary tries to ensure that the resulting corrupted distribution under either hypotheses is the same, rendering those sample useless for hypothesis testing.

If the corruption rate is greater than $\frac{1}{2}$ for a sample, then the adversary can trivially 'simulate' the distribution $\frac{1}{2} \left( \mathcal{N}(\delta, 1) + \mathcal{N}(-\delta, 1) \right)$. Alternately, when $\delta$ is small, adversary can simulate the distribution $\max\{\mathcal{N}(\delta, 1), \mathcal{N}(-\delta, 1)\}/Z(\delta)$ instead, where $Z(\delta)$ is an appropriate normalizing constant. The adversary can simulate this distribution when the corruption rate is greater than $1 - e^{-\delta}$. Thus, only the samples in the set $\{i : \lambda_i < \min\{\frac{1}{2}, 1 - e^{-\delta}\}\}$ are relevant for hypothesis testing, and it remains to choose a value of $\delta$ judiciously.

Based on our generic multivariate bounds of Theorem 4 later, we have the following Corollary 1 for the univariate case.

**Corollary 1** (Minimax Rate for Univariate Gaussian Distributions)**.** *Suppose $\min_{t \in [0,1]} \left( t^2 + \frac{1}{N(t)} \right) \leq c$ for some universal constants $c$, then*

$$L_{\mathsf{PAC}}(\boldsymbol{\lambda}, \mathcal{D}_1^{\mathcal{N}}) \simeq \min_{t \in [0,1]} \left( \frac{1}{N(t)} + t^2 \right). \tag{22}$$

*Moreover, the estimator $\hat{\mu}_{\mathsf{S}}(\mathbf{Z}, w(t^*))$, with $t^* = \arg\min_{t \in [0,1]} \left( t^2 + \frac{1}{N(t)} \right)$, achieves this error.*

Thus the proposed weighted Tukey median scheme is optimal in the univariate case – demonstrating that beyond a certain corruption threshold determined by $N(\cdot)$, there is no way to leverage the more contaminated samples up to universal constants.

**Multivariate Case** In the multivariate setting, our upper and lower bounds (see Appendix C.2) are off by a multiplicative factor of $\sqrt{d}$. This gap can possibly be attributed by our rather relaxed handling of the heterogeneity since the proof technique we consider can recover the minimax rate of $\frac{d}{n} + \epsilon^2$ in the homogeneous setting where $\lambda_i = \epsilon \; \forall i$ (see Appendix F).

**Theorem 4.** *Suppose* $\min_{t \in [0,1]} \left( t^2 + \frac{d}{N(t)} \right) \leq c$ *for some universal constants c, then*

$$\frac{1}{\sqrt{d}} \min_{t \in [0,1]} \left( \frac{d}{N(t)} + t^2 \right) \lesssim L_{\mathsf{PAC}}(\boldsymbol{\lambda}, \mathcal{D}_1^{\mathcal{N}}) \lesssim \min_{t \in [0,1]} \left( \frac{d}{N(t)} + t^2 \right). \tag{23}$$

*Moreover, the estimator* $\hat{\mu}_{\mathsf{S}}(\boldsymbol{Z}, w(t^*))$, *with* $t^* = \arg\min_{t \in [0,1]} \left( t^2 + \frac{d}{N(t)} \right)$, *achieves the upper bound.*

The proof of Theorem 4 can be found in Appendix C.3. As Theorem 4 suggests, it might be possible to leverage the heterogeneity beyond just discarding samples with a higher threshold and this might reduce the squared-error by a factor of $\sqrt{d}$. However, as stated before, this gap might be an artifact of the analysis rather than a true phenomenon. We leave this as an open question.

## 4 Linear Regression

We now consider the problem of linear regression under Gaussian covariates. The proofs for this section can be found in Appendix D.

**Background:** Under no corruptions, the minimax rate is known to be $\frac{\sigma^2 d}{n}$, i.e., independent of $\Sigma$ as long as $\Sigma$ is non-singular – the lack of information in directions with low eigenvalues of $\Sigma$ if offset by the $\| \cdot \|_\Sigma$ distance. In the homogeneous robust regression setting with $\epsilon$ corruption, the minimax rate is of the order $\frac{\sigma^2 d}{n} + \sigma^2 \epsilon^2$ [31].

For the upper bound, we consider a weighted version of Tukey median adapted to regression [31, 42]. Define the weighted regression depth of a point $\eta \in \mathbb{R}^d$ for a dataset $\boldsymbol{Z}$ with $Z_i = (\hat{W}_i, \hat{Y}_i)$ and $w \in \Delta_n$ as

$$D_w(\eta, \boldsymbol{Z}) = \min_{v \in \mathbb{S}_d} \sum_{i=1}^n w_i \mathbb{I}\{(\hat{Y}_i - \eta^T \hat{W}_i)(v^T \hat{W}_i) \geq 0\}. \tag{24}$$

The regression depth of a point $\eta \in \mathbb{R}^d$ is high if for all half-spaces, the sign of the residual errors using $\eta$ are distributed equally in every direction. The weighted Tukey regression coefficient is defined as

$$\hat{\beta}_{\mathsf{TC}}(\boldsymbol{Z}, w) := \arg\max_{\eta \in \mathbb{R}^d} D_w(\eta, \boldsymbol{Z}). \tag{25}$$

**Proposition 2** (Upper Bound for Regression). *For all* $w \in \Delta_n$ *satisfying* $(w^T \lambda)^2 + d\|w\|_2^2 \leq c$ *for some universal constants c,*

$$\sup_{P \in \mathcal{D}(\Sigma, \sigma^2)} \Pr_{\boldsymbol{Z} \sim \boldsymbol{\lambda} P} \left[ \|\hat{\beta}_{\mathsf{TC}}(\boldsymbol{Z}, w) - \beta_P\|_\Sigma^2 \geq c' \left( (w^T \lambda)^2 + d\|w\|_2^2 \right) \right] \leq 1/5, \tag{26}$$

*for some universal constant* $c'$.

Like (20), we can similarly consider a thresholding method – define $\hat{\beta}_{\mathsf{S}}(\boldsymbol{Z}, t) = \hat{\beta}_{\mathsf{TM}}(\boldsymbol{Z}, w(t))$. The lower bound construction is similar in spirit to Theorem 4. Based on our upper and lower bounds, we present Theorem 5.

**Theorem 5** (Minimax Rate for Linear Regression). *Suppose* $\min_{t \in [0,1]} \left( t^2 + \frac{d}{N(t)} \right) \leq c$ *for some universal constants c, then*

$$\frac{\sigma^2}{\sqrt{d}} \min_{t \in [0,1]} \left( \frac{d}{N(t)} + t^2 \right) \lesssim L_{\mathsf{reg}}(\boldsymbol{\lambda}, \mathcal{D}(\Sigma, \sigma^2)) \lesssim \sigma^2 \min_{t \in [0,1]} \left( \frac{d}{N(t)} + t^2 \right). \tag{27}$$

*Moreover, the estimator* $\hat{\beta}_{\mathsf{S}}(\boldsymbol{Z}, w(t^*))$, *with* $t^* = \arg\min_{t \in [0,1]} \left( t^2 + \frac{d}{N(t)} \right)$, *achieves the upper bound.*

Note that when $\lambda_i = \epsilon \; \forall i$, we can recover the homogeneous minimax rate of $\frac{\sigma^2 d}{n} + \sigma^2 \epsilon^2$ using the more refined lower bound in Appendix F.

# 5 Discussion and Future Work

There exists several avenues of future work. Perhaps most pressing issue would be to address the $\sqrt{d}$ gap in our upper and lower bounds for multivariate Gaussian mean estimation and linear regression. In this regard, it seems that neither our upper nor or lower bound in Theorem 4 can be tight for all choices of $\lambda$. Indeed, the lower bound in Equation (23) is not tight for homogeneous corruption (although this can be avoided with our more sophisticated lower bound technique, see Appendix F), and on the other hand, the upper bound fails to be tight in the aforementioned regime of semi-verified learning, where the upper bound would suggest that the error in the estimator should diverge as $d \to \infty$, whereas existing upper bounds due to [7, 9] already demonstrate that dimension independent rates are possible. This suggests that fundamentally new measures of robustness are needed to fully characterize the complexity of learning under heterogeneous errors in the high-dimensional setting. The heterogeneous robust estimation problem can also be studied under different corruption models such as total-variation distance or Kullback-Leibler divergence corruption.

## Acknowledgments and Disclosure of Funding

S.C. acknowledges the support of AI Policy Hub, UC Berkeley.

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

Figure 2: Mean estimation algorithms for (a) bounded distributions and (b) univariate Gaussian distributions. The x-axis is a proxy for degree of contamination of the model.

# A  Experiments

We investigate the practical performance of our algorithms and compare it with baseline in Figure 2. For bounded mean estimation, we use the sample mean as the baseline since it is the minimax robust estimator in the homogeneous setting. For univariate Gaussian mean estimation, we use the sample median, equivalent to Tukey median in one dimension, as the baseline for the same reason.

We set $n = 10^4$ and for a fixed value $q$, we sample the corruption rates $\boldsymbol{\lambda}$ i.i.d. from the distribution with cdf given by $F(t) = 1 - (1-t)^q$. As $q$ increases we can expect a higher corruption rate. Fixing this sampled $\boldsymbol{\lambda}$, we sample the dataset $10^4$ times. For bounded distribution, we plot the mean squared-error and the corresponding standard deviations over the trials at each value of $q$ considered. For the Gaussian distribution, we plot the empirical $\frac{4}{5}$-th quantile of the squared-error along with $\frac{15}{20}$-th and $\frac{17}{20}$-th quantiles over the trials. For the bounded distribution, we choose $r = 1$ and choose the true underlying distribution to be the point mass at $0$, and the corrupted values to be $1$. For univariate Gaussian distribution, we fix the true distribution to be $\mathcal{N}(0, 1)$ and the corrupted values sampled i.i.d. from $\mathcal{N}(100, 1)$.

Optimal linear method in the plots refer to the reweighing scheme proposed while threshold method refers to the special case of reweighing that discards samples above a certain corruption threshold and performs standard homogeneous robust estimation on the sub-sampled dataset.

While unclear for the Gaussian distribution, the reweighing does seem to provide marginal improvement over thresholding method. Further investigation is required to establish whether reweighing may pose significant advantages in high dimensions.

# B  Bounded Mean Estimation: Proofs

## B.1  Variance Upper Bound

Using $E\|X + Y\|_2^2 \leq 2E\|X\|_2^2 + 2E\|Y\|_2^2$, we get

$$\mathbb{E}\left[\left\|\sum_{i=1}^n w_i(Z_i - \mathbb{E}[Z_i])\right\|_2^2\right] = \mathbb{E}\left[\left\|\sum_{i=1}^n w_i((1 - B_i)X_i - (1 - \lambda_i)\mu_P) + \sum_{i=1}^n w_i(B_i\tilde{X}_i - \lambda_i\mu_{Q_i})\right\|_2^2\right]$$
(28)

$$\leq 2\mathbb{E}\left[\left\|\sum_{i=1}^n w_i((1 - B_i)X_i - (1 - \lambda_i)\mu_P)\right\|_2^2\right] + 2\mathbb{E}\left[\left\|\sum_{i=1}^n w_i(B_i\tilde{X}_i - \lambda_i\mu_{Q_i})\right\|_2^2\right]$$
(29)

Since $\{(1-B_i)X_i - (1-\lambda_i)\mu_P\}$ are independent random variables and $\|(1-B_i)X_i\|_2 \le r$, we can use the crude variance bound $E\|(1-B_i)X_i - (1-\lambda_i)\mu_P\|_2^2 \le r^2$ to obtain

$$\mathbb{E}\left[\left\|\sum_{i=1}^n w_i((1-B_i)(X_i - \mu_P) - (1-\lambda_i)\mu_P)\right\|_2^2\right] \le r^2\|w\|_2^2. \tag{30}$$

Inspecting the other term, we use the law of total variance by conditioning on $B$. In particular,

$$\mathbb{E}\left[\left\|\sum_{i=1}^n w_i(B_i\tilde{X}_i - \lambda_i\mu_{Q_i})\right\|_2^2\right] = \mathbb{E}\left[\left\|\sum_{i=1}^n w_i(B_i(\tilde{X}_i - \mu_{Q_i}) + \mu_{Q_i}(B_i - \lambda_i))\right\|_2^2\right] \tag{31}$$

$$\le 2\mathbb{E}\left[\left\|\sum_{i=1}^n w_iB_i(\tilde{X}_i - \mu_{Q_i})\right\|_2^2\right] + 2\mathbb{E}\left[\left\|\sum_i w_i\mu_{Q_i}(B_i - \lambda_i))\right\|_2^2\right]. \tag{32}$$

For the first term in (32), use Jensen's inequality as

$$\mathbb{E}\left[\left\|\sum_{i=1}^n w_iB_i(\tilde{X}_i - \mu_{Q_i})\right\|_2^2\right] = \mathbb{E}\left[\mathbb{E}\left[\left\|\sum_{i=1}^n w_iB_i(\tilde{X}_i - \mu_{Q_i})\right\|_2^2\,\Big|\,B\right]\right] \tag{33}$$

$$\le 4r^2\mathbb{E}\left[\left(\sum_{i=1}^n w_iB_i\right)^2\right] \tag{34}$$

$$= 4r^2\sum_i w_i^2\lambda_i(1-\lambda_i) + 4r^2(w^T\lambda)^2 \tag{35}$$

$$\le r^2\|w\|_2^2 + 4r^2(w^T\lambda)^2. \tag{36}$$

For the second term in (32), using the fact that $\|\mu_{Q_i}\|_2 \le r$, using Jensen's inequality we get

$$\mathbb{E}\left[\left\|\sum_i w_i\mu_{Q_i}(B_i - \lambda_i))\right\|_2^2\right] \le r^2\mathbb{E}\left[\left(\sum_i w_i(B_i - \lambda_i))\right)^2\right] \le \frac{r^2}{4}\|w\|_2^2. \tag{37}$$

Combining the above, obtain

$$\mathbb{E}\left[\left\|\sum_{i=1}^n w_i(Z_i - \mathbb{E}[Z_i])\right\|_2^2\right] \le 7r^2\|w\|_2^2 + 16r^2(w^T\lambda)^2 \tag{38}$$

## B.2 Upper Bound Solution

To solve

$$\min_{w\in\Delta_n} \|w\|^2 + c(w^T\lambda)^2, \tag{39}$$

consider the Lagrangian $\mathcal{L}(w,\beta,\gamma) = \|w\|_2^2 + c(w^T\lambda)^2 + 2\beta(1 - \sum_i w_i) - \sum_i 2\gamma_i w_i$. KKT condition on the Lagrangian leads to

$$w_i = \beta - c(w^T\lambda)\lambda_i + \gamma_i \;\forall i, \tag{40}$$

where $\gamma_i w_i = 0, \gamma_i \ge 0 \;\forall i$. Thus, we can equivalently write

$$w_i = (\beta - c(w^T\lambda)\lambda_i)_+. \tag{41}$$

Notice that $w_i$ are decreasing in $\lambda_i$ thus, order the indices such that $\lambda_1 \le \lambda_2 \ldots \le \lambda_n$. Note that since this is a strictly convex objective with a convex compact constraint set we are guaranteed a *unique* solution $w$.

Let $m$ such that $w_i > 0 \ \forall i \leq m$ and $w_i = 0 \ \forall i > m$; if no such $m$ exists then it is understood $m = n$. Since $\sum w_i = 1$, use (41) to obtain the condition

$$m\beta - c(w^T\lambda)\|\lambda_1^m\|_1 = 1. \tag{42}$$

Noting that $w^T\lambda = \sum_{i=1}^m w_i\lambda_i$, we can use (41) to obtain

$$w^T\lambda = \beta\|\lambda_1^m\|_1 - cw^T\lambda\|\lambda_1^m\|_2^2. \tag{43}$$

Solving for $w^T\lambda$ and substituting in (42), obtain $\beta = \frac{1+c\|\lambda_1^m\|_2^2}{k(1+c\|\lambda_1^m\|_2^2)-c\|\lambda_1^m\|_1^2}$, and

$$w_i = \frac{1+c\|\lambda_1^m\|_2^2}{m(1+c\|\lambda_1^m\|_2^2) - c\|\lambda_1^m\|_1^2}\left(1 - c\lambda_i\frac{\|\lambda_1^m\|_1}{1+c\|\lambda_1^m\|_2^2}\right)_+ \ \forall i \in [n]. \tag{44}$$

Thus, the problem of solving for the weights has been reduced to identifying the index $k$ after which the weights are zero. This is precisely what Algorithm 1 does. In particular, $m + 1 = \min\{j : w_j = 0\}$ by definition and $w_{m+1} = 0 \Leftrightarrow \lambda_{m+1} \geq \frac{1+c\|\lambda_1^m\|_2^2}{c\|\lambda_1^m\|_1}$ by (44). Therefore, if the loop in Algorithm 1 runs without termination till index $k = m$, then it will correctly terminate at $k = m$ since $\lambda_{m+1} \geq \frac{1+c\|\lambda_1^m\|_2^2}{c\|\lambda_1^m\|_1}$.

Thus, we need to show that the algorithm does not terminate before $k = m$ to prove correctness. Assume the contrary that it terminates at $k = p < m$, i.e., $\lambda_{p+1} \geq \frac{1+c\|\lambda_1^p\|_2^2}{c\|\lambda_1^p\|_1}$. Observe that

$$\lambda_{p+1} \geq \frac{1+c\|\lambda_1^p\|_2^2}{c\|\lambda_1^p\|_1} \Leftrightarrow \lambda_{p+1} \geq \frac{1+c\|\lambda_1^{p+1}\|_2^2}{c\|\lambda_1^{p+1}\|_1} \tag{45}$$

$$\implies \lambda_{p+2} \geq \frac{1+c\|\lambda_1^{p+1}\|_2^2}{c\|\lambda_1^{p+1}\|_1}, \tag{46}$$

where (46) follows since $\lambda$(s) are indexed in non-decreasing order. Extending this argument, we get $\lambda_m \geq \frac{1+c\|\lambda_1^m\|_2^2}{c\|\lambda_1^m\|_1}$. Note since $w_m > 0$, we have

$$\lambda_m < \frac{1+c\|\lambda_1^m\|_2^2}{c\|\lambda_1^m\|_1} \tag{47}$$

by (44) – a contradiction. This proves that the proposed algorithm solves for $w$ correctly.

## B.3 Lower Bound

By Le Cam's method,

$$L(\lambda, r) \geq r^2\delta^2\left(1 - \mathsf{TV}\left(\otimes_{i=1}^n \mathsf{Ber}\left(\frac{1}{2} - \epsilon_i\right), \otimes_{i=1}^n \mathsf{Ber}\left(\frac{1}{2} + \epsilon_i\right)\right)\right), \tag{48}$$

$$= r^2\delta^2\left(1 - \sqrt{\frac{1}{2}\sum_{i=1}^n \mathsf{KL}\left(\mathsf{Ber}\left(\frac{1}{2} - \epsilon_i\right), \mathsf{Ber}\left(\frac{1}{2} + \epsilon_i\right)\right)}\right) \tag{49}$$

$$= r^2\delta^2\left(1 - \sqrt{6\sum_{i=1}^n \epsilon_i^2}\right), \tag{50}$$

where we used $2\epsilon\log\frac{1+2\epsilon}{1-2\epsilon} \leq 12\epsilon^2 \ \forall \epsilon \in [0, \frac{1}{4}]$. Let $n(t) = |\{i : \lambda_i < t\}|$, then we obtain

$$L(\lambda, r) \geq r^2\delta^2\left(1 - \sqrt{6\delta^2 n\left(\frac{2\delta}{1+2\delta}\right)}\right) \tag{51}$$

$$\geq r^2\delta^2\left(1 - \sqrt{6\delta^2 n(2\delta)}\right) \ \forall \delta \in \left[0, \frac{1}{4}\right] \tag{52}$$

## C    Mean Estimation for Gaussian Distributions: Proofs

### C.1    Upper Bound

Recall

$$D_w(\eta, \boldsymbol{Z}) = \min_{v \in \mathbb{S}_d} \sum_{i=1}^{n} w_i \mathbb{I}\{v^T(Z_i - \eta) \geq 0\}, \tag{53}$$

$$\hat{\mu}_{\mathsf{TM}}(\boldsymbol{Z}, w) := \arg\max_{\eta \in \mathbb{R}^d} D_w(\eta, \boldsymbol{Z}). \tag{54}$$

Let $G = \{i : B_i = 0\}$ and $B = [n] \setminus G$. Note that $Z_i = X_i$ for $i \in G$. The depth of the true mean is lower bounded as

$$D_w(\mu, \boldsymbol{Z}) \geq \min_{v \in \mathbb{S}_d} \sum_{i \in G} w_i \mathbb{I}\{(X_i - \mu)^T v \geq 0\}. \tag{55}$$

Define the class of indicator functions $\mathcal{F}_\mu = \{f_v(x) = \mathbb{I}\{(x - \mu)^T v \geq 0\} | v \in \mathbb{S}_d\}$. Note that $E[f(X)] = \frac{1}{2} \; \forall f \in \mathcal{F}_\mu$. With some abuse of notation, let $w(G) = \{w_i | i \in G\}$. By Proposition 4, we have with probability at least $1 - \frac{\delta}{4}$

$$\min_{f \in \mathcal{F}_\mu} \sum_{i \in G} w_i \left( f(X_i) - \frac{1}{2} \right) \geq -62 \|w(G)\|_2 \sqrt{\mathsf{VC}(\mathcal{F}_\mu)} - \|w(G)\|_2 \sqrt{\frac{\log 4/\delta}{2}} \tag{56}$$

$$\geq -\|w\|_2 \left( 62\sqrt{d} + \sqrt{\frac{\log 4/\delta}{2}} \right), \tag{57}$$

where we used $\mathsf{VC}(\mathcal{F}_\mu) = d$; readers may refer to [43, Corollary 4.2.2] for VC dimension of homogeneous half-space classifiers. Further, with probability at least $1 - \delta/4$, by McDiarmid's inequality [28]

$$\sum_{i \in G} w_i = \sum_{i=1}^{n} w_i \mathbb{I}\{B_i = 0\} \tag{58}$$

$$\geq \sum_{i=1}^{n} w_i (1 - \lambda_i) - \|w\|_2 \sqrt{\frac{\log 4/\delta}{2}} \tag{59}$$

$$= 1 - w^T \lambda - \|w\|_2 \sqrt{\frac{\log 4/\delta}{2}}. \tag{60}$$

Thus, with probability at least $1 - \delta/2$,

$$D_w(\mu, \boldsymbol{Z}) \geq \frac{1}{2} - \frac{w^T \lambda}{2} - \|w\|_2 \left( 62\sqrt{d} + \sqrt{\frac{9 \log 4/\delta}{8}} \right). \tag{61}$$

Next, we show that depth of any point far away from the true mean is low. For any $\eta \in \mathbb{R}^d$ such that $\|\eta - \mu\|_2 \geq r = \Phi^{-1}(\frac{1}{2} + \alpha)$, let $v_\eta = \frac{\eta - \mu}{\|\eta - \mu\|_2}$. We shall set the value of $\alpha > 0$ later.

$$\sup_{\eta : \|\eta - \mu\|_2 \geq r} D_w(\eta, \boldsymbol{Z}) \leq \sum_{i \in B} w_i + \sup_{\eta : \|\eta - \mu\|_2 \geq r} \sum_{i=1}^{n} w_i \mathbb{I}\{(X_i - \eta)^T v_\eta \geq 0\}. \tag{62}$$

Define the class of indicator functions $\mathcal{G}_\mu = \{f_\eta(x) = \mathbb{I}\{(x - \eta)^T v_\eta \geq 0\} | \|\eta - \mu\|_2 \geq r\}$. Since $\mathbb{E}[\mathbb{I}\{(X - \eta)^T v_\eta \geq 0\}] = \Phi(-\|\eta - \mu\|_2)$, we have $E[f(X)] \leq \frac{1}{2} - \alpha \; \forall f \in \mathcal{G}_\mu$.

Now, note that $\mathcal{G}_\mu \subseteq \{f_\eta(x) = \mathbb{I}\{(x - \eta)^T v_\eta \geq 0\} | \eta \in \mathbb{R}^d\}$. By reparameterzing $x$, we have $\mathsf{VC}(\{f_\eta(x) = \mathbb{I}\{(x - \eta)^T v_\eta \geq 0\} | \eta \in \mathbb{R}^d\}) = \mathsf{VC}(\{f_\eta(x) = \mathbb{I}\{(x - \eta)^T \eta \geq 0\} | \eta \in \mathbb{R}^d\})$.

Observe that $\mathsf{VC}(\{f_\eta(x) = \mathbb{I}\{(x - \eta)^T \eta \geq 0\}|\eta \in \mathbb{R}^d\}) \leq \mathsf{VC}(\{f_{\eta,v}(x) = \mathbb{I}\{(x - \eta)^T v \geq 0\}|\eta, v \in \mathbb{R}^d\}) = d + 1$. Thus, $\mathsf{VC}(\mathcal{G}_\mu) \leq d + 1 \leq 2d$.

Thus, by Proposition 4, with probability at least $1 - \delta/4$,

$$\sup_{\eta:\|\eta-\mu\|_2 \geq r} D_w(\eta, \boldsymbol{Z}) \leq \sum_{i \in B} w_i + \sup_{g \in \mathcal{G}_\mu} \sum_{i=1}^{n} w_i g(X_i) \tag{63}$$

$$\leq \sum_{i \in B} w_i + \frac{1}{2} - \alpha + \|w\|_2 \left(62\sqrt{2d} + \sqrt{\frac{\log 4/\delta}{2}}\right). \tag{64}$$

Again, by McDiarmid's inequality, with probability at least $1 - \delta/4$,

$$\sum_{i \in B} w_i \leq w^T\lambda + \|w\|_2\sqrt{\frac{\log 4/\delta}{2}}. \tag{65}$$

Thus, with probability at least $1 - \delta/2$, we have

$$\sup_{\eta:\|\eta-\mu\|_2 \geq r} D_w(\eta, \boldsymbol{Z}) \leq \frac{1}{2} - \alpha + w^T\lambda + \|w\|_2 \left(88\sqrt{d} + 2\sqrt{\frac{\log 4/\delta}{2}}\right). \tag{66}$$

Combining (61) and (66), picking $\alpha = \frac{3}{2}w^T\lambda + \|w\|_2 \left(150\sqrt{d} + 3.5\sqrt{\frac{\log 4/\delta}{2}}\right)$ ensures that no point $\eta$ such that $\|\mu - \eta\|_2 \geq \Phi^{-1}(\frac{1}{2} + \alpha)$ can be returned by the Tukey median estimator. Thus, with probability at least $1 - \delta$, we have

$$\|\hat{\mu}_{\mathsf{TM}} - \mu\|_2 \leq \Phi^{-1}\left(\frac{1}{2} + \alpha\right) \tag{67}$$

$$\leq 3\alpha, \tag{68}$$

using the identity $\Phi^{-1}\left(\frac{1}{2} + x\right) \leq 3x \; \forall x \in [0, \frac{1}{3}]$. The above upper bound is valid as long as $\alpha = \frac{3}{2}w^T\lambda + \|w\|_2 \left(150\sqrt{d} + 3.5\sqrt{\frac{\log 4/\delta}{2}}\right) < \frac{1}{3}$.

Setting $\delta = 1/5$, ensuring $\frac{3}{2}w^T\lambda + \|w\|_2 \left(150\sqrt{d} + 4.3\right) < \frac{3}{2}w^T\lambda + 155\|w\|_2\sqrt{d} \leq \frac{1}{3}$ suffices.

Thus, summarizing, let $g(w, \lambda) = \frac{3}{2}w^T\lambda + 155\|w\|_2\sqrt{d}$. For $w$ and $\lambda$ such that $g(w, \lambda) \leq \frac{1}{3}$, we have the guarantee that with probability at least $\frac{4}{5}$,

$$\|\hat{\mu}_{\mathsf{TM}} - \mu\|_2 \leq 3g(w, \lambda). \tag{69}$$

To reduce the above condition to the simpler form stated in the main paper, note that

$$g(w, \lambda)^2 \leq \left(\frac{3}{2}w^T\lambda + 155\|w\|_2\sqrt{d}\right)^2 \tag{70}$$

$$\leq \left(155w^T\lambda + 155\|w\|_2\sqrt{d}\right)^2 \tag{71}$$

$$\leq 2 \times 155^2 \left((w^T\lambda)^2 + d\|w\|_2^2\right). \tag{72}$$

Ensuring the above is less than $\frac{1}{9}$ suffices. Thus, $\forall w \in \Delta_n$ such that $(w^T\lambda)^2 + d\|w\|_2^2 \leq \frac{1}{432450}$, we have

$$\sup_{P \in \mathcal{D}_d^N} \Pr_{\boldsymbol{Z} \sim_\lambda P} \left[\|\hat{\mu}_{\mathsf{TM}}(\boldsymbol{Z}, w) - \mu_P\|_2^2 \geq 432450 \left((w^T\lambda)^2 + d\|w\|_2^2\right)\right] \leq 1/5 \tag{73}$$

Correspondingly, the threshold based estimator satisfies the following – $\forall t \in [0, 1]$ such that $t^2 + \frac{d}{N(t)} \leq \frac{1}{432450}$, we have

$$\sup_{P \in \mathcal{D}_d^N} \Pr_{\boldsymbol{Z} \sim_\lambda P} \left[\|\hat{\mu}_{\mathsf{S}}(\boldsymbol{Z}, w(t)) - \mu_P\|_2^2 \geq 432450 \left(t^2 + \frac{d}{N(t)}\right)\right] \leq 1/5 \tag{74}$$

## C.2 Lower Bound

We provide a lower bound for Gaussian mean estimation in $\mathbb{R}^d$ in this section. For a vector $v \in \mathbb{R}^{\sqrt{d}}$, let $e(v)$ denote a vector in $\mathbb{R}^d$ with $v$ as the first $\sqrt{d}$ elements and the last $d - \sqrt{d}$ elements equal to 0. Define the parameterized distribution $P_\delta(\tau) = \mathcal{N}(\delta e(\tau), I)$ for $\delta > 0$ to be specified later, and let $\mathcal{P}_\delta = \{P_\delta(\tau) | \tau \in \{-1, 1\}^{\sqrt{d}}\}$.

**Note:** The supremum in our minimax definition is over both the true distribution and the adversarial strategy. We shall specify the adversarial strategy later for our lower bound argument.

Note that $L_{\mathsf{PAC}}(\boldsymbol{\lambda}, \mathcal{D}_d^{\mathcal{N}}) \geq L_{\mathsf{PAC}}(\boldsymbol{\lambda}, \mathcal{P}_\delta) \ \forall \delta$ and thus, we shall lower bound $L_{\mathsf{PAC}}(\boldsymbol{\lambda}, \mathcal{P}_\delta)$.

We begin by lower bounding

$$L_{\mathbb{E}}(\boldsymbol{\lambda}, \mathcal{P}_\delta) = \inf_M \sup_{P \in \mathcal{P}_\delta} \mathbb{E}_{\boldsymbol{Z} \sim_\lambda P} \left[ \|M(\boldsymbol{Z}) - \mu_P\|_2^2 \right]. \tag{75}$$

For a vector $\tau$, let $\tau'^j$ denote the vector such that $\tau_i = \tau_i'^j \forall i \neq j$ and $\tau_j = -\tau_j'^j$. Using Assouad's lower bound technique [28],

$$\inf_M \sup_{P \in \mathcal{P}_\delta} \mathbb{E}_{\boldsymbol{Z} \sim_\lambda P} \left[ \|M(\boldsymbol{Z}) - \mu_P\|_2^2 \right] \geq \inf_M \frac{1}{2^{\sqrt{d}}} \sum_{\tau \in \{-1,1\}^{\sqrt{d}}} \mathbb{E} \left[ \|M(\boldsymbol{Z}) - \delta e(\tau)\|_2^2 \right] \tag{76}$$

$$= \inf_M \frac{1}{2^{\sqrt{d}}} \sum_{\tau \in \{-1,1\}^{\sqrt{d}}} \sum_{j=1}^d \mathbb{E} \left[ |M(\boldsymbol{Z})_j - \delta e(\tau)_j|^2 \right] \tag{77}$$

$$= \inf_M \sum_{j=1}^d \frac{1}{2^{\sqrt{d}}} \sum_{\tau \in \{-1,1\}^{\sqrt{d}}} \mathbb{E} \left[ |M(\boldsymbol{Z})_j - \delta e(\tau)_j|^2 \right] \tag{78}$$

$$\geq \inf_M \sum_{j=1}^{\sqrt{d}} \frac{1}{2^{\sqrt{d}}} \sum_{\tau \in \{-1,1\}^{\sqrt{d}}} \mathbb{E} \left[ |M(\boldsymbol{Z})_j - \delta \tau_j|^2 \right]. \tag{79}$$

Now, note that

$$\sum_{\tau \in \{-1,1\}^{\sqrt{d}}} \mathbb{E} \left[ |M(\boldsymbol{Z})_j - \delta \tau_j|^2 \right] = \sum_{\tau \in \{-1,1\}^{\sqrt{d}}} \frac{\mathbb{E} \left[ |M(\boldsymbol{Z})_j - \delta \tau_j|^2 \right] + \mathbb{E} \left[ |M(\boldsymbol{Z})_j - \delta \tau_j'^j|^2 \right]}{2} \ \forall j \tag{80}$$

Thus, we get

$$L_{\mathbb{E}}(\boldsymbol{\lambda}, \mathcal{P}_\delta) \geq \inf_M \sum_{j=1}^{\sqrt{d}} \frac{1}{2^{\sqrt{d}}} \sum_{\tau \in \{-1,1\}^{\sqrt{d}}} \frac{\mathbb{E} \left[ |M(\boldsymbol{Z})_j - \delta \tau_j|^2 \right] + \mathbb{E} \left[ |M(\boldsymbol{Z})_j - \delta \tau_j'^j|^2 \right]}{2} \tag{81}$$

$$\geq \sum_{j=1}^{\sqrt{d}} \frac{1}{2^{\sqrt{d}}} \sum_{\tau \in \{-1,1\}^{\sqrt{d}}} \inf_M \frac{\mathbb{E} \left[ |M(\boldsymbol{Z})_j - \delta \tau_j|^2 \right] + \mathbb{E} \left[ |M(\boldsymbol{Z})_j - \delta \tau_j'^j|^2 \right]}{2} \tag{82}$$

$$\geq \sum_{j=1}^{\sqrt{d}} \frac{1}{2^{\sqrt{d}}} \sum_{\tau \in \{-1,1\}^{\sqrt{d}}} \delta^2 (1 - \mathsf{TV}(P_{\boldsymbol{Z} \sim_\lambda P_\delta(\tau)}, P_{\boldsymbol{Z} \sim_\lambda P_\delta(\tau'^j)})), \tag{83}$$

where the last line follows by Le Cam's method and the notation $P_{\boldsymbol{Z} \sim_\lambda P_\delta(\tau)}$ refers to the distribution of $\boldsymbol{Z}$ when the true distribution is $P_\delta(\tau)$. We shall now specify a particular adversarial strategy to lower bound the above.

**Adversarial strategy motivation:** Consider a particular sample with corruption rate $\lambda$ and let the underlying true distribution be $P_\delta(\tau)$. Denote the perturbed sample as $Z(\tau)$ and the outlier to be $\tilde{X}(\tau)$. Note that $Z(\tau) \sim (1 - \lambda) P_{X(\tau)} + \lambda P_{\tilde{X}(\tau)}$. For any particular clean sample $X(\tau) \sim P_\delta(\tau)$, the adversary's goal is to ensure the sample $Z(\tau)$ contains no information about $\tau$. One possible way

is that the adversary can try to ensure $Z(\tau) \sim \frac{\max_{\tau'} P(\tau')}{T}$, where the maximum is taken pointwise over the pdf and $T$ is a normalizing constant. Observe the identity

$$P(\tau) + \left(\max_{\tau' \neq \tau} P(\tau') - P(\tau)\right)_+ = \max_{\tau'} P(\tau'), \tag{84}$$

where $(\cdot)_+ := \max\{\cdot, 0\}$ is pointwise over the pdf. First, note that

$$\int_{x \in \mathbb{R}^d} \left(\max_{\tau' \neq \tau} P_\delta(\tau')(x) - P_\delta(\tau)(x)\right)_+ dx = T - 1,$$

where $P_\delta(\tau)(x)$ is understood to be the pdf of $P_\delta(\tau)$ at $x$. Thus, $\frac{\left(\max_{\tau' \neq \tau} P(\tau') - P(\tau)\right)_+}{T-1}$ is a valid pdf, and for $\lambda = \frac{T-1}{T}$, we have $(1-\lambda)P(\tau) + \lambda \frac{\left(\max_{\tau' \neq \tau} P(\tau') - P(\tau)\right)_+}{T-1} = \frac{\max_{\tau'} P(\tau')}{T}$. Thus, for any $\lambda \geq 1 - \frac{1}{T}$, there exists a way for the adversary to make the distribution of $Z(\tau) \sim \frac{\max_{\tau'} P(\tau')}{T}$, rendering the sample useless for identifying $\tau$.

Now, we find an upper bound on $T$ in terms of $\delta$.

$$T = \int_{x \in \mathbb{R}^d} \max_{\tau \in \{-1,1\}^{\sqrt{d}}} \frac{1}{\sqrt{(2\pi)^d}} e^{-\frac{\|x - \delta e(\tau)\|_2^2}{2}} dx \tag{85}$$

$$= \int_{x \in \mathbb{R}^d} \frac{1}{\sqrt{(2\pi)^d}} \max_{\tau \in \{-1,1\}^{\sqrt{d}}} e^{-\frac{\|x - \delta e(\tau)\|_2^2}{2}} dx \tag{86}$$

$$= \int_{x' \in \mathbb{R}^{\sqrt{d}}} \frac{1}{\sqrt{(2\pi)^{\sqrt{d}}}} \max_{\tau \in \{-1,1\}^{\sqrt{d}}} e^{-\frac{\|x' - \delta\tau\|_2^2}{2}} dx' \tag{87}$$

$$= \left[\int_{x'' \in \mathbb{R}} \frac{1}{\sqrt{(2\pi)}} \max_{\tau \in \{-1,1\}} e^{-\frac{\|x'' - \delta\tau\|_2^2}{2}} dx''\right]^{\sqrt{d}} \tag{88}$$

$$= \left[2 \int_{y \in \mathbb{R}: y \geq 0} \frac{1}{\sqrt{2\pi}} e^{-\frac{\|y - \delta\|_2^2}{2}} dy\right]^{\sqrt{d}} \tag{89}$$

$$= [2\Phi(\delta)]^{\sqrt{d}}, \tag{90}$$

where $\Phi(\cdot)$ is the cumulative distribution function of standard normal distribution. Using $2\Phi(x) \leq 1 + x$ and $1 + x \leq e^x$, we obtain

$$T \leq e^{\delta\sqrt{d}}. \tag{91}$$

Thus, if $\lambda \geq 1 - e^{-\delta\sqrt{d}}$ then the adversary can ensure that $Z(\tau) \sim \frac{\max_{\tau'} P(\tau')}{T}$ and contains no information about $\tau$.

**Adversarial strategy:** If for sample $i$, $\lambda_i \geq 1 - e^{-\delta\sqrt{d}}$, then independently sample $\tilde{X}(\tau) \sim \beta \left(\frac{\max_{\tau' \neq \tau} P_\delta(\tau') - P_\delta(\tau)}{T-1}\right)_+ + (1-\beta)\frac{\max_{\tau'} P_\delta(\tau')}{T}$, where $\beta = \frac{(T-1)(1-\lambda)}{\lambda}$. The constraint $\lambda_i \geq 1 - e^{-\delta\sqrt{d}}$ ensures $\beta \in [0,1]$ and the above is a valid distribution.

Thus, $Z(\tau) \sim \frac{\max_{\tau'} P_\delta(\tau')}{T}$. If $\lambda_i < 1 - e^{-\delta\sqrt{d}}$, then adversary does no corruption, or equivalently, independently sample $\tilde{X}(\tau) \sim P_\delta(\tau)$ so that $Z(\tau) \sim P_\delta(\tau)$.

Thus, using Pinsker's inequality in (83), obtain

$$L_{\mathbb{E}}(\boldsymbol{\lambda}, \mathcal{P}_\delta) \geq \sum_{j=1}^{\sqrt{d}} \frac{1}{2\sqrt{d}} \sum_{\tau \in \{-1,1\}^{\sqrt{d}}} \delta^2 \left(1 - \sqrt{\frac{1}{2}\mathsf{KL}(P_{\boldsymbol{Z} \sim_\lambda P_\delta(\tau)}, P_{\boldsymbol{Z} \sim_\lambda P_\delta(\tau'^j)})}\right) \tag{92}$$

$$= \sum_{j=1}^{\sqrt{d}} \frac{1}{2\sqrt{d}} \sum_{\tau \in \{-1,1\}^{\sqrt{d}}} \delta^2 \left(1 - \sqrt{\frac{1}{2}\mathsf{KL}\left(\otimes_{i=1}^n P_{Z_i(\tau)}, \otimes_{i=1}^n P_{Z_i(\tau'^j)}\right)}\right) \tag{93}$$

$$= \sum_{j=1}^{\sqrt{d}} \frac{1}{2\sqrt{d}} \sum_{\tau \in \{-1,1\}^{\sqrt{d}}} \delta^2 \left(1 - \sqrt{\frac{1}{2} \sum_{i:\lambda_i < 1-e^{-\delta\sqrt{d}}} \mathsf{KL}\left(P_\delta(\tau), P_\delta(\tau'^j)\right)}\right) \tag{94}$$

$$= \sqrt{d}\delta^2 \left(1 - \sqrt{\delta^2 n(1 - e^{-\delta\sqrt{d}})}\right) \quad \forall \delta \tag{95}$$

Let $\delta_*$ be such that

$$n(1 - e^{-\delta_*\sqrt{d}}) \leq \frac{1}{64\delta_*^2}, \quad \text{and } N(1 - e^{-\delta_*\sqrt{d}}) \geq \frac{1}{64\delta_*^2}. \tag{96}$$

Substituting $\delta_*$ in (95),

$$\frac{7}{8}\sqrt{d}\delta_*^2 \leq \inf_M \sup_{P \in \mathcal{P}_{\delta_*}} \mathbb{E}_{\boldsymbol{Z} \sim_\lambda P}\left[\|M(\boldsymbol{Z}) - \mu_P\|_2^2\right]. \tag{97}$$

For a random variable $K$, let $Q(K, \alpha) := \inf\{t : \Pr[K \geq t] \leq (1-\alpha)\}$, i.e., $Q(\cdot, \alpha)$ is the $\alpha$-quantile of the random variable. Note that for a random variable $K$, $\mathbb{E}K \leq Q(K, x)x + (1-x)\text{ess-sup}K$ $\forall x \in [0,1]$, where ess-sup denotes essential supremum. Further, note that in the minimax term $\inf_M \sup_{P \in \mathcal{P}_{\delta_*}}$, we can restrict ourselves to estimators $M$ which output in $[-\delta_*, \delta_*]^{\sqrt{d}} \times \{0\}^{d-\sqrt{d}}$ almost surely – otherwise the error can be reduced by projected to this region. Thus ess-sup$\|M(\boldsymbol{Z}) - \mu_P\|_2^2 \leq 4\sqrt{d}\delta_*^2$ for any estimator $M$ and any distribution $P$. Thus, combining the above observations, we have

$$\inf_M \sup_{P \in \mathcal{P}_{\delta_*}} \mathbb{E}_{\boldsymbol{Z} \sim_\lambda P}\|M(\boldsymbol{Z}) - \mu_P\|_2^2 \leq \inf_M \sup_{P \in \mathcal{P}_{\delta_*}} \frac{4}{5}Q\left(\|M(\boldsymbol{Z}) - \mu_P\|_2^2, \frac{4}{5}\right) + \frac{4\sqrt{d}\delta_*^2}{5}. \tag{98}$$

Using (97), we get

$$\frac{7}{8}\sqrt{d}\delta_*^2 \leq \frac{4}{5}\inf_M \sup_{P \in \mathcal{P}_{\delta_*}} Q\left(\|M(\boldsymbol{Z}) - \mu_P\|_2^2, \frac{4}{5}\right) + \frac{4\sqrt{d}\delta_*^2}{5} \tag{99}$$

$$= \frac{4}{5}L_{\mathsf{PAC}}(\boldsymbol{\lambda}, \mathcal{P}_{\delta_*}) + \frac{4\sqrt{d}\delta_*^2}{5} \tag{100}$$

$$\implies \frac{3}{40}\sqrt{d}\delta_*^2 \leq L_{\mathsf{PAC}}(\boldsymbol{\lambda}, \mathcal{P}_{\delta_*}) \leq L_{\mathsf{PAC}}(\boldsymbol{\lambda}, \mathcal{D}_d^{\mathcal{N}}). \tag{101}$$

## C.3 Minimax Optimality

For proving Theorem 4, we shall use the weighing $w_i = \frac{\mathbb{I}\{\lambda_i \leq 1-e^{\delta_*\sqrt{d}}\}}{N(1-e^{\delta_*\sqrt{d}})}$ in our upper bound. From (74), $\forall t \in [0,1]$ such that $t^2 + \frac{d}{N(t)} \leq c$ for some universal constant $c$, we have

$$\|\hat{\mu}_{\mathsf{S}}(\boldsymbol{Z}, w(t)) - \mu\|_2^2 \lesssim \left(t^2 + \frac{d}{N(t)}\right). \tag{102}$$

Thus, if $\min_{t \in [0,1]} \left( t^2 + \frac{d}{N(t)} \right) \leq c$ and let the minimum value be attained at $t^*$, then we have

$$\| \hat{\mu}_{\mathsf{S}}(\mathbf{Z}, w(t^*)) - \mu \|_2^2 \lesssim \left( t^{*2} + \frac{d}{N(t^*)} \right) \tag{103}$$

$$\leq \left( 1 - e^{-\delta_* \sqrt{d}} \right)^2 + \frac{d}{N \left( 1 - e^{-\delta_* \sqrt{d}} \right)} \tag{104}$$

$$\leq d\delta_*^2 + 64 d\delta_*^2 \tag{105}$$

$$\simeq d\delta_*^2, \tag{106}$$

where we used (96) in (105). Combining with (101), when $\min_{t \in [0,1]} \left( t^2 + \frac{d}{N(t)} \right) \leq c$, we have

$$\sqrt{d}\delta_*^2 \lesssim L_{\mathsf{PAC}}(\boldsymbol{\lambda}, \mathcal{D}_d^{\mathcal{N}}) \lesssim \min_{t \in [0,1]} \left( t^2 + \frac{d}{N(t)} \right) \leq d\delta_*^2. \tag{107}$$

Thus, when $\min_{t \in [0,1]} \left( t^2 + \frac{d}{N(t)} \right) \leq c$,

$$\frac{1}{\sqrt{d}} \min_{t \in [0,1]} \left( t^2 + \frac{d}{N(t)} \right) \lesssim L_{\mathsf{PAC}}(\boldsymbol{\lambda}, \mathcal{D}_d^{\mathcal{N}}) \lesssim \min_{t \in [0,1]} \left( t^2 + \frac{d}{N(t)} \right). \tag{108}$$

## D   Linear Regression: Proofs

### D.1   Upper Bound

Recall

$$D_w(\eta, \mathbf{Z}) = \min_{v \in \mathbb{S}_d} \sum_{i=1}^{n} w_i \mathbb{I}\{ (\hat{Y}_i - \eta^T \hat{W}_i)(v^T \hat{W}_i) \geq 0 \}, \tag{109}$$

$$\hat{\beta}_{\mathsf{TC}}(\mathbf{Z}, w) := \arg\max_{\eta \in \mathbb{R}^d} D_w(\eta, \mathbf{Z}). \tag{110}$$

Let the true underlying regression coefficient be $\beta$, i.e., $W \sim \mathcal{N}(0, \Sigma)$ and conditioned on $W$, $Y \sim \mathcal{N}(\beta^T W, \sigma^2)$. Let $G = \{i : B_i = 0\}$, $B = [n] \setminus G$, and Let $w(G) = \{w_i | i \in G\}$. Note that $Z_i = (W_i, Y_i)$ for $i \in G$. The depth of the true coefficient is lower bounded as

$$D_w(\beta, \mathbf{Z}) \geq \min_{v \in \mathbb{S}_d} \sum_{i \in G} w_i \mathbb{I}\{ (Y_i - \beta^T W_i)(v^T W_i) \geq 0 \}. \tag{111}$$

Define the class of indicator functions $\mathcal{F}_\beta = \{ f_v(w, y) = \mathbb{I}\{ (y - w^T \beta)(v^T w) \geq 0 \} | v \in \mathbb{S}_d \}$. Note that $E[f(Z)] = \frac{1}{2} \ \forall f \in \mathcal{F}_\beta$. By Proposition 4, we have with probability at least $1 - \frac{\delta}{4}$

$$\min_{f \in \mathcal{F}_\beta} \sum_{i \in G} w_i \left( f(X_i) - \frac{1}{2} \right) \geq -62 \| w(G) \|_2 \sqrt{\mathsf{VC}(\mathcal{F}_\beta)} - \| w(G) \|_2 \sqrt{\frac{\log 4/\delta}{2}} \tag{112}$$

$$\geq -\| w \|_2 \left( 62\sqrt{d} + \sqrt{\frac{\log 4/\delta}{2}} \right), \tag{113}$$

where we used $\mathsf{VC}(\mathcal{F}_\beta) = d$. To see this, notice $\mathbb{I}\{ (y - w^T \beta)(v^T w) \geq 0 \} = \mathbb{I}\{ v^T (w(y - w^T \beta)) \geq 0 \} = \mathbb{I}\{ v^T \tilde{w} \geq 0 \}$, where $\tilde{w} = w(y - w^T \beta) \in \mathbb{R}^d$. Thus, it is equal to the VC dimension of homogeneous half-space classifiers, which is $d$ [43, Corollary 4.2.2]. Further, with probability at

least $1 - \delta/4$, by McDiarmid's inequality [28]

$$\sum_{i \in G} w_i = \sum_{i=1}^{n} w_i \mathbb{I}\{B_i = 0\} \tag{114}$$

$$\geq \sum_{i=1}^{n} w_i(1 - \lambda_i) - \|w\|_2 \sqrt{\frac{\log 4/\delta}{2}} \tag{115}$$

$$= 1 - w^T\lambda - \|w\|_2 \sqrt{\frac{\log 4/\delta}{2}}. \tag{116}$$

Thus, with probability at least $1 - \delta/2$,

$$D_w(\beta, \boldsymbol{Z}) \geq \frac{1}{2} - \frac{w^T\lambda}{2} - \|w\|_2 \left(62\sqrt{d} + 1.5\sqrt{\frac{\log 4/\delta}{2}}\right). \tag{117}$$

Next, we show that depth of any point far away from the coefficient is low. For any $\eta \in \mathbb{R}^d$ such that $\|\eta - \mu\|_\Sigma \geq r$, let $v_\eta = \frac{\eta - \beta}{\|\eta - \mu\|_2}$. We shall set the value of $r > 0$ later.

$$\sup_{\eta : \|\eta - \mu\|_2 \geq r} D_w(\eta, \boldsymbol{Z}) \leq \sum_{i \in B} w_i + \sup_{\eta : \|\eta - \mu\|_2 \geq r} \sum_{i=1}^{n} w_i \mathbb{I}\{(Y_i - \eta^T W_i)v_\eta^T W_i \geq 0\}. \tag{118}$$

Again, by McDiarmid's inequality, with probability at least $1 - \delta/4$,

$$\sum_{i \in B} w_i \leq w^T\lambda + \|w\|_2 \sqrt{\frac{\log 4/\delta}{2}}. \tag{119}$$

Define the class of indicator functions $\mathcal{G}_\beta = \{f_\eta(w, y) = \mathbb{I}\{(y - \eta^T w)(v_\eta^T w) \geq 0\}|\|\eta - \mu\|_\Sigma \geq r\}$. Thus, by Proposition 4, with probability at least $1 - \delta/4$,

$$\sup_{f \in \mathcal{G}_\beta} \sum_{i=1}^{n} w_i(f(Z_i) - \mathbb{E}[f(Z_i)]) \leq 62\|w\|_2 \sqrt{\mathsf{VC}(\mathcal{G}_\beta)} + \|w\|_2 \sqrt{\frac{\log 4/\delta}{2}} \tag{120}$$

Now, note that

$$\mathbb{E}[f(Z)] = \Pr\left[(Y - \eta^T W)W^T(\eta - \beta) \geq 0\right] \tag{121}$$

Using $W \sim \mathcal{N}(0, \Sigma)$ and $Y|W \sim \mathcal{N}(\beta^T W, \sigma^2)$, we get that

$$(Y - \eta^T W)W^T(\eta - \beta) \sim M(\zeta - M) \tag{122}$$

where $M = W^T(\eta - \beta) \sim \mathcal{N}(0, \|\eta - \beta\|_\Sigma^2)$ and $\zeta \sim \mathcal{N}(0, \sigma^2)$ are independent. Letting $T_1, T_2 \sim_{iid} \mathcal{N}(0, 1)$,

$$\Pr\left[(Y - \eta^T W)W^T(\eta - \beta) \geq 0\right] = \Pr\left[M(\zeta - M) \geq 0\right] \tag{123}$$

$$= \Pr\left[\zeta \geq M | M \geq 0\right] \tag{124}$$

$$= \Pr\left[\sigma T_2 \geq \|\eta - \beta\|_\Sigma T_1 | T_1 \geq 0\right] \tag{125}$$

$$= 1 - \Pr\left[T_2 \leq \frac{\|\eta - \beta\|_\Sigma}{\sigma} T_1 | T_1 \geq 0\right] \tag{126}$$

$$= 1 - \left(\frac{1}{2} + \frac{1}{\pi}\arctan\frac{\|\eta - \beta\|_\Sigma}{\sigma}\right) \tag{127}$$

$$= \frac{1}{2} - \frac{1}{\pi}\arctan\frac{\|\eta - \beta\|_\Sigma}{\sigma} \tag{128}$$

Thus,

$$\mathbb{E}[f(Z)] \leq \frac{1}{2} - \frac{1}{\pi}\arctan\frac{r}{\sigma} \quad \forall f \in \mathcal{G}_\beta. \tag{129}$$

Thus, with probability at least $1 - \delta/2$, and using VC dimension bound presented in Proposition 3,

$$\sup_{\eta:\|\eta-\mu\|_\Sigma \geq r} D_w(\eta, \mathbf{Z}) \leq \frac{1}{2} - \frac{1}{\pi}\arctan\frac{r}{\sigma} + w^T\lambda + \|w\|_2\left(2\sqrt{\frac{\log 4/\delta}{2}} + 879\sqrt{d}\right). \quad (130)$$

Combining with (117), setting

$$r = \sigma\tan\left\{\pi\left[\frac{3}{2}w^T\lambda + \|w\|_2\left(3.5\sqrt{\frac{\log 4/\delta}{2}} + 941\sqrt{d}\right)\right]\right\} \quad (131)$$

ensures that the estimator $\hat{\beta}_{\mathsf{TC}}$ satisfies

$$\|\hat{\beta}_{\mathsf{TC}} - \beta\|_\Sigma \leq r \quad (132)$$

with probability at least $1 - \delta$ as long as $\left[\frac{3}{2}w^T\lambda + \|w\|_2\left(3.5\sqrt{\frac{\log 4/\delta}{2}} + 941\sqrt{d}\right)\right] \leq \frac{2}{5}$.

Substitute $\delta = \frac{1}{5}$ and let $g(w, \lambda) = \frac{3}{2}w^T\lambda + 946\|w\|_2\sqrt{d}$. Then, for $w$ such that $g(w, \lambda) \leq \frac{2}{5}$, we have with probability at least $\frac{4}{5}$,

$$\|\hat{\beta}_{\mathsf{TC}} - \beta\|_\Sigma \leq \sigma\tan\pi g(w, \lambda) \leq 8\sigma g(w, \lambda), \quad (133)$$

where we used the identity $\tan\pi x \leq 8x \ \forall x \in [0, 2/5]$.

To reduce the above condition to the simpler form stated in the main paper, note that

$$\left(\frac{3}{2}w^T\lambda + 946\|w\|_2\sqrt{d}\right)^2 \leq \left(946w^T\lambda + 946\|w\|_2\sqrt{d}\right)^2 \quad (134)$$

$$\leq 2 \times 946^2\left((w^T\lambda)^2 + d\|w\|_2^2\right). \quad (135)$$

Ensuring the above is less than $\frac{4}{25}$ suffices.

Thus, $\forall w \in \Delta_n$ such that $(w^T\lambda)^2 + d\|w\|_2^2 \leq \frac{1}{11186450}$, we have

$$\sup_{P \in \mathcal{D}_d^\mathcal{N}} \Pr_{\mathbf{Z} \sim \lambda P}\left[\|\hat{\beta}_{\mathsf{TC}}(\mathbf{Z}, w) - \mu_P\|_\Sigma^2 \geq 114549248\sigma^2\left((w^T\lambda)^2 + d\|w\|_2^2\right)\right] \leq 1/5 \quad (136)$$

Correspondingly, the threshold based estimator satisfies the following – $\forall t \in [0, 1]$ such that $t^2 + \frac{d}{N(t)} \leq \frac{1}{11186450}$, we have

$$\sup_{P \in \mathcal{D}_d^\mathcal{N}} \Pr_{\mathbf{Z} \sim \lambda P}\left[\|\hat{\mu}_{\mathsf{S}}(\mathbf{Z}, w(t)) - \mu_P\|_\Sigma^2 \geq 114549248\sigma^2\left(t^2 + \frac{d}{N(t)}\right)\right] \leq 1/5 \quad (137)$$

**Proposition 3.** $\mathsf{VC}(\mathcal{G}_\beta) \leq 200d$.

*Proof.* Recall $\mathcal{G}_\beta = \{f_\eta(w, y) = \mathbb{I}\{(y - \eta^T w)(v_\eta^T w) \geq 0\}\|\|\eta - \mu\|_\Sigma \geq r\}$, where $v_\eta = \frac{\eta - \beta}{\|\eta - \beta\|_2}$. Let $\mathcal{G}_\beta^+ = \{f_\eta(w, y) = \mathbb{I}\{(y - \eta^T w)(v_\eta^T w) \geq 0\}|\eta \in \mathbb{R}^d, \eta \neq \mu\}$. Note that for the purposes of VC dimension calculation, by re-parameterizing $y$ and $\eta$, we can write

$$\mathcal{G}_\beta^+ = \{g_\eta(w, y) = \mathbb{I}\{(y - \eta^T w)(\eta^T w) \geq 0\}|\eta \in \mathbb{R}^d, \eta \neq 0\}. \quad (138)$$

We now switch to region notation instead of a functional notation. Let $G_\eta = \{(w, y)|g_\eta(w, y) = 1\}$ and define the following

- $H_\eta^{1+} = \{(w, y)|y - \eta^T w \geq 0\}$,

- $H_\eta^{1-} = \{(w, y)|y - \eta^T w \leq 0\}$,

- $H_\eta^{2+} = \{(w, y)|\eta^T w \geq 0\}$,

- $H_\eta^{2-} = \{(w, y) | \eta^T w \leq 0\}$.

Note that $G_\eta = \left(H_\eta^{1+} \cap H_\eta^{2+}\right) \cup \left(H_\eta^{1-} \cap H_\eta^{2-}\right)$. Define $\mathcal{G} = \{G_\eta | \eta \in \mathbb{R}^d, \eta \neq 0\}$, $\mathcal{H}^+ = \{H_\eta^{1+} \cap H_\eta^{2+} | \eta \in \mathbb{R}^d, \eta \neq 0\}$ and $\mathcal{H}^- = \{H_\eta^{1-} \cap H_\eta^{2-} | \eta \in \mathbb{R}^d, \eta \neq 0\}$.

We use the following property related to VC dimension [43]:

$$\mathsf{VC}(\{A \cap B | A \in \mathcal{A}, B \in \mathcal{B}\}) \leq 10 \max\{\mathsf{VC}(\mathcal{A}), \mathsf{VC}(\mathcal{B})\}. \tag{139}$$

The above holds for union as well.

Noting that $\mathcal{G} \subseteq \{G = H_1 \cup H_2 | H_1 \in \mathcal{H}^+, H_2 \in \mathcal{H}^-\}$. Define $T = \max\{\mathsf{VC}(\mathcal{H}^+), \mathsf{VC}(\mathcal{H}^-)\}$ then $\mathsf{VC}(\mathcal{G}_\beta^+) = \mathsf{VC}(\mathcal{G}) \leq 10T$. Similarly, by writing $\mathcal{H}^+ \subseteq \{H_\eta^{1+} \cap H_{\eta'}^{2+} | \eta, \eta' \in \mathbb{R}^d, \eta, \eta' \neq 0\}$, we have $\mathsf{VC}(\mathcal{H}^+) \leq 10 \max\{\mathsf{VC}(\{H_\eta^{1+}\}), \mathsf{VC}(\{H_\eta^{2+}\})\} = 10(d+1)$, where we used $\mathsf{VC}(\{H_\eta^{1+}\}) = d + 1$ and $\mathsf{VC}(\{H_\eta^{2+}\}) = d$. Similarly, $\mathsf{VC}(\mathcal{H}^-) \leq 10(d+1)$.

Thus, $\mathsf{VC}(\mathcal{G}_\beta) \leq 100(d+1) \leq 200d$. $\qquad\square$

## D.2 Lower Bound

The lower bound argument we present is similar to Appendix C.2.

We begin by noting that when true regression coefficient is $\beta$ then $(W, Y) \sim \mathcal{N}\left(0, \begin{bmatrix} \Sigma & \Sigma\beta \\ \beta^T\Sigma & \beta^T\Sigma\beta + \sigma^2 \end{bmatrix}\right)$.

For a vector $v \in \mathbb{R}^{\sqrt{d}}$, let $e(v)$ denote a vector in $\mathbb{R}^d$ with $v$ as the first $\sqrt{d}$ elements and the last $d - \sqrt{d}$ elements equal to 0. Let $\beta(\tau) = \delta\Sigma^{-\frac{1}{2}}e(\tau)$ for $\delta > 0$ to be specified later. For $\tau \in \{-1, 1\}^{\sqrt{d}}$, define the $Y | W \sim \mathcal{N}(W^T\beta(\tau), \sigma^2)$. In other words, for $\tau \in \{-1, 1\}^{\sqrt{d}}$, $(W, Y) \sim P_\delta(\tau)$ where

$$P_\delta(\tau) = \mathcal{N}\left(0, \begin{bmatrix} \Sigma & \Sigma\beta(\tau) \\ \beta(\tau)^T\Sigma & \beta(\tau)^T\Sigma\beta(\tau) + \sigma^2 \end{bmatrix}\right)$$

and let $\mathcal{P}_\delta = \{P_\delta(\tau) | \tau \in \{-1, 1\}^{\sqrt{d}}\}$.

We have $L_{\text{reg}}(\boldsymbol{\lambda}, \mathcal{D}(\Sigma, \sigma^2)) \geq L_{\text{reg}}(\boldsymbol{\lambda}, \mathcal{P}_\delta) \; \forall \delta$ and thus, we shall lower bound $L_{\text{reg}}(\boldsymbol{\lambda}, \mathcal{P}_\delta)$.

We begin by lower bounding

$$L_{\mathbb{E}}(\boldsymbol{\lambda}, \mathcal{P}_\delta) = \inf_M \sup_{P \in \mathcal{P}_\delta} \mathbb{E}_{\boldsymbol{Z} \sim_\lambda P} \left[\|M(\boldsymbol{Z}) - \beta_P\|_\Sigma^2\right]. \tag{140}$$

For a vector $\tau$, let $\tau'^j$ denote the vector such that $\tau_i = \tau_i'^j \forall i \neq j$ and $\tau_j = -\tau_j'^j$. For an estimation $M(\boldsymbol{Z})$, define $N(\boldsymbol{Z}) = \Sigma^{-\frac{1}{2}}M(\boldsymbol{Z})$ – note that this is a bijective map. Using Assouad's lower bound

technique [28],

$$\inf_M \sup_{P \in \mathcal{P}_\delta} \mathbb{E}_{\boldsymbol{Z} \sim_\lambda P} \left[ \|M(\boldsymbol{Z}) - \beta_P\|_\Sigma^2 \right] \geq \inf_M \frac{1}{2^{\sqrt{d}}} \sum_{\tau \in \{-1,1\}^{\sqrt{d}}} \mathbb{E} \left[ \|M(\boldsymbol{Z}) - \beta(\tau)\|_\Sigma^2 \right] \tag{141}$$

$$= \inf_M \frac{1}{2^{\sqrt{d}}} \sum_{\tau \in \{-1,1\}^{\sqrt{d}}} \mathbb{E} \left[ \|N(\boldsymbol{Z}) - \delta e(\tau)\|_2^2 \right] \tag{142}$$

$$= \inf_N \frac{1}{2^{\sqrt{d}}} \sum_{\tau \in \{-1,1\}^{\sqrt{d}}} \mathbb{E} \left[ \|N(\boldsymbol{Z}) - \delta e(\tau)\|_2^2 \right] \tag{143}$$

$$= \inf_N \frac{1}{2^{\sqrt{d}}} \sum_{\tau \in \{-1,1\}^{\sqrt{d}}} \sum_{j=1}^d \mathbb{E} \left[ |N(\boldsymbol{Z})_j - \delta e(\tau)_j|^2 \right] \tag{144}$$

$$= \inf_N \sum_{j=1}^d \frac{1}{2^{\sqrt{d}}} \sum_{\tau \in \{-1,1\}^{\sqrt{d}}} \mathbb{E} \left[ |N(\boldsymbol{Z})_j - \delta e(\tau)_j|^2 \right] \tag{145}$$

$$\geq \inf_N \sum_{j=1}^{\sqrt{d}} \frac{1}{2^{\sqrt{d}}} \sum_{\tau \in \{-1,1\}^{\sqrt{d}}} \mathbb{E} \left[ |N(\boldsymbol{Z})_j - \delta \tau_j|^2 \right]. \tag{146}$$

Now, note that

$$\sum_{\tau \in \{-1,1\}^{\sqrt{d}}} \mathbb{E} \left[ |N(\boldsymbol{Z})_j - \delta\tau_j|^2 \right] = \sum_{\tau \in \{-1,1\}^{\sqrt{d}}} \frac{\mathbb{E} \left[ |N(\boldsymbol{Z})_j - \delta\tau_j|^2 \right] + \mathbb{E} \left[ |N(\boldsymbol{Z})_j - \delta\tau_j'^j|^2 \right]}{2} \ \forall j \tag{147}$$

Thus, we get

$$L_\mathbb{E}(\boldsymbol{\lambda}, \mathcal{P}_\delta) \geq \inf_N \sum_{j=1}^{\sqrt{d}} \frac{1}{2^{\sqrt{d}}} \sum_{\tau \in \{-1,1\}^{\sqrt{d}}} \frac{\mathbb{E} \left[ |N(\boldsymbol{Z})_j - \delta\tau_j|^2 \right] + \mathbb{E} \left[ |N(\boldsymbol{Z})_j - \delta\tau_j'^j|^2 \right]}{2} \tag{148}$$

$$\geq \sum_{j=1}^{\sqrt{d}} \frac{1}{2^{\sqrt{d}}} \sum_{\tau \in \{-1,1\}^{\sqrt{d}}} \inf_N \frac{\mathbb{E} \left[ |N(\boldsymbol{Z})_j - \delta\tau_j|^2 \right] + \mathbb{E} \left[ |N(\boldsymbol{Z})_j - \delta\tau_j'^j|^2 \right]}{2} \tag{149}$$

$$\geq \sum_{j=1}^{\sqrt{d}} \frac{1}{2^{\sqrt{d}}} \sum_{\tau \in \{-1,1\}^{\sqrt{d}}} \delta^2 (1 - \mathsf{TV}(P_{\boldsymbol{Z} \sim_\lambda P_\delta(\tau)}, P_{\boldsymbol{Z} \sim_\lambda P_\delta(\tau'^j)})), \tag{150}$$

where the last line follows by Le Cam's method and the notation $P_{\boldsymbol{Z} \sim_\lambda P_\delta(\tau)}$ refers to the distribution of $\boldsymbol{Z}$ when the true distribution is $P_\delta(\tau)$. We shall now specify a particular adversarial strategy to lower bound the above.

**Adversarial strategy motivation:** Readers can refer to the motivation in Appendix C.2 for more details on the main idea behind the strategy described. We need to find an upper bound on the normalizing constant $T$ for the pdf $\frac{\max_\tau P_\delta(\tau)}{T}$. Let

$$S(\tau) = \begin{bmatrix} \Sigma & \Sigma\beta(\tau) \\ \beta(\tau)^T\Sigma & \beta(\tau)^T\Sigma\beta(\tau) + \sigma^2 \end{bmatrix}$$

Thus,

$$T = \int_{w \in \mathbb{R}^d, y \in \mathbb{R}} \max_{\tau \in \{-1,1\}^{\sqrt{d}}} \frac{1}{\sqrt{(2\pi)^{d+1}|S(\tau)|}} e^{-\frac{\|(w,y)\|_{S(\tau)^{-1}}^2}{2}} \, dw dy \tag{151}$$

By Schur's formula, $|S(\tau)| = |\Sigma|\sigma^2$ and by block inversion formula,

$$S(\tau)^{-1} = \begin{bmatrix} \Sigma^{-1} + \frac{\beta(\tau)\beta(\tau)^T}{\sigma^2} & -\frac{\beta(\tau)}{\sigma^2} \\ -\frac{\beta(\tau)^T}{\sigma^2} & \frac{1}{\sigma^2} \end{bmatrix}.$$

Performing change of variable $(w, y) = G(x, y)$ where $G = \begin{bmatrix} \Sigma^{\frac{1}{2}} & 0 \\ 0 & 1 \end{bmatrix}$, we obtain

$$T = \int_{x \in \mathbb{R}^d, y \in \mathbb{R}} \max_{\tau \in \{-1,1\}^{\sqrt{d}}} \frac{1}{\sqrt{(2\pi)^{d+1}|\Sigma|\sigma^2}} e^{-\frac{(x,y)^T G^T S(\tau)^{-1} G(x,y)}{2}} \begin{vmatrix} \Sigma^{\frac{1}{2}} & 0 \\ 0 & 1 \end{vmatrix} dx dy \qquad (152)$$

Noting that

$$G^T S(\tau)^{-1} G = \begin{bmatrix} 1 + \frac{\delta^2}{\sigma^2} e(\tau)e(\tau)^T & -\frac{\delta}{\sigma^2} e(\tau) \\ -\frac{\delta}{\sigma^2} e(\tau)^T & \frac{1}{\sigma^2} \end{bmatrix},$$

we get

$$T = \int_{x \in \mathbb{R}^d, y \in \mathbb{R}} \max_{\tau \in \{-1,1\}^{\sqrt{d}}} \frac{1}{\sqrt{(2\pi)^{d+1}\sigma^2}} e^{-\frac{\|x\|_2^2}{2} - \frac{(\delta e(\tau)^T x - y)^2}{2\sigma^2}} dx dy \qquad (153)$$

$$= \int_{x \in \mathbb{R}^d} \frac{1}{\sqrt{(2\pi)^d}} e^{-\|x\|_2^2/2} \int_{y \in \mathbb{R}} \frac{1}{\sqrt{2\pi\sigma^2}} \max_{\tau \in \{-1,1\}^{\sqrt{d}}} e^{-\frac{(\delta e(\tau)^T x - y)^2}{2\sigma^2}} dy dx \qquad (154)$$

$$= \int_{x \in \mathbb{R}^{\sqrt{d}}} \frac{1}{\sqrt{(2\pi)^{\sqrt{d}}}} e^{-\|x\|_2^2/2} \int_{y \in \mathbb{R}} \frac{1}{\sqrt{2\pi\sigma^2}} \max_{\tau \in \{-1,1\}^{\sqrt{d}}} e^{-\frac{(\delta \tau^T x - y)^2}{2\sigma^2}} dy dx \qquad (155)$$

Let $X \sim \mathcal{N}(0, I)$ be a random variable in $\mathbb{R}^{\sqrt{d}}$. We can write

$$T = \mathbb{E}_X \left[ \int_{y \in \mathbb{R}} \frac{1}{\sqrt{2\pi\sigma^2}} \max_{\tau \in \{-1,1\}^{\sqrt{d}}} e^{-\frac{(\delta \tau^T X - y)^2}{2\sigma^2}} dy \right] \qquad (156)$$

Note that $\max_{\tau \in \{-1,1\}^{\sqrt{d}}} \delta \tau^T X = \delta \|X\|_1$. Thus,

$$\max_{\tau \in \{-1,1\}^{\sqrt{d}}} e^{-\frac{(\delta \tau^T X - y)^2}{2\sigma^2}} \begin{cases} = e^{-(y - \delta\|X\|_1)^2/2\sigma^2} & y \geq \delta\|X\|_1, \\ = e^{-(y + \delta\|X\|_1)^2/2\sigma^2} & y \leq -\delta\|X\|_1, \\ \leq 1 & \text{else.} \end{cases}$$

Using the above upper bound, we obtain

$$T \leq \mathbb{E}_X \left[ 1 + \frac{2\delta\|X\|_1}{\sqrt{2\pi\sigma^2}} \right] \qquad (157)$$

$$= 1 + \frac{2\delta\mathbb{E}[\|X\|_1]}{\sqrt{2\pi\sigma^2}} \qquad (158)$$

$$= 1 + \frac{2\delta\sqrt{d}}{\pi\sigma} \qquad (159)$$

$$\leq e^{\frac{2\delta\sqrt{d}}{\pi\sigma}}. \qquad (160)$$

Thus, if $\lambda \geq 1 - e^{-\frac{2\delta\sqrt{d}}{\pi\sigma}}$ then the adversary can ensure that $Z(\tau) \sim \frac{\max_{\tau'} P(\tau')}{T}$ and contains no information about $\tau$.

**Adversarial strategy:** If for sample $i$, $\lambda_i \geq 1 - e^{-\frac{2\delta\sqrt{d}}{\pi\sigma}}$, then independently sample $\tilde{X}(\tau) \sim \beta \left( \frac{\max_{\tau' \neq \tau} P_\delta(\tau') - P_\delta(\tau)}{T-1} \right)_+ + (1 - \beta)\frac{\max_{\tau'} P_\delta(\tau')}{T}$, where $\beta = \frac{(T-1)(1-\lambda)}{\lambda}$. The constraint $\lambda_i \geq 1 - e^{-\frac{2\delta\sqrt{d}}{\pi\sigma}}$ ensures $\beta \in [0, 1]$ and the above is a valid distribution.

Thus, $Z(\tau) \sim \frac{\max_{\tau'} P_\delta(\tau')}{T}$. If $\lambda_i < 1 - e^{-\frac{2\delta\sqrt{d}}{\pi\sigma}}$, then adversary does no corruption, or equivalently, independently sample $\tilde{X}(\tau) \sim P_\delta(\tau)$ so that $Z(\tau) \sim P_\delta(\tau)$.

Thus, using Pinsker's inequality in (150), obtain

$$L_{\mathbb{E}}(\boldsymbol{\lambda}, \mathcal{P}_\delta) \geq \sum_{j=1}^{\sqrt{d}} \frac{1}{2\sqrt{d}} \sum_{\tau \in \{-1,1\}^{\sqrt{d}}} \delta^2 \left( 1 - \sqrt{\frac{1}{2} \mathsf{KL}(P_{\boldsymbol{Z} \sim_\lambda P_\delta(\tau)}, P_{\boldsymbol{Z} \sim_\lambda P_\delta(\tau'^j)})} \right) \tag{161}$$

$$= \sum_{j=1}^{\sqrt{d}} \frac{1}{2\sqrt{d}} \sum_{\tau \in \{-1,1\}^{\sqrt{d}}} \delta^2 \left( 1 - \sqrt{\frac{1}{2} \mathsf{KL}\left( \otimes_{i=1}^n P_{Z_i(\tau)}, \otimes_{i=1}^n P_{Z_i(\tau'^j)} \right)} \right) \tag{162}$$

$$= \sum_{j=1}^{\sqrt{d}} \frac{1}{2\sqrt{d}} \sum_{\tau \in \{-1,1\}^{\sqrt{d}}} \delta^2 \left( 1 - \sqrt{\frac{1}{2} \sum_{i:\lambda_i < 1 - e^{-\frac{2\delta\sqrt{d}}{\pi\sigma}}} \mathsf{KL}\left( P_\delta(\tau), P_\delta(\tau'^j) \right)} \right) \tag{163}$$

$$= \sqrt{d}\delta^2 \left( 1 - \sqrt{\frac{\delta^2}{\sigma^2} n(1 - e^{-\frac{2\delta\sqrt{d}}{\pi\sigma}})} \right) \quad \forall \delta \tag{164}$$

where (164) follows since

$$\mathsf{KL}\left( P_\delta(\tau), P_\delta(\tau'^j) \right) = \mathbb{E}_{W \sim \mathcal{N}(0,\Sigma)} \mathsf{KL}\left( \mathcal{N}(W^T \beta(\tau), \sigma^2), \mathcal{N}(W^T \beta(\tau'^j), \sigma^2) \right) \tag{165}$$

$$= \mathbb{E}_W \frac{\delta^2 (\tau - \tau'^j)^T \Sigma^{-\frac{1}{2}} W W^T \Sigma^{-\frac{1}{2}} (\tau - \tau'^j)}{2\sigma^2} \tag{166}$$

$$= \frac{\delta^2 \|\tau - \tau'^j\|_2^2}{2\sigma^2} \tag{167}$$

$$= \frac{2\delta^2}{\sigma^2}. \tag{168}$$

Replacing $\delta \leftarrow \delta/\sigma$ in (164), we get

$$L_{\mathbb{E}}(\boldsymbol{\lambda}, \mathcal{P}_{\sigma\delta}) \geq \sigma^2 \sqrt{d}\delta^2 \left( 1 - \sqrt{\delta^2 n(1 - e^{-\frac{2\delta\sqrt{d}}{\pi}})} \right) \quad \forall \delta, \tag{169}$$

Let $\delta_*$ be such that

$$n(1 - e^{-\frac{2\delta_* \sqrt{d}}{\pi}}) \leq \frac{1}{64\delta_*^2}, \quad \text{and } N(1 - e^{-\frac{2\delta_* \sqrt{d}}{\pi}}) \geq \frac{1}{64\delta_*^2}. \tag{170}$$

Substituting $\delta_*$ in (169),

$$\frac{7}{8} \sqrt{d}\sigma^2 \delta_*^2 \leq \inf_M \sup_{P \in \mathcal{P}_{\sigma\delta_*}} \mathbb{E}_{\boldsymbol{Z} \sim_\lambda P} \left[ \|M(\boldsymbol{Z}) - \beta_P\|_\Sigma^2 \right]. \tag{171}$$

Note that in the minimax term $\inf_M \sup_{P \in \mathcal{P}_{\delta_*}}$, we can restrict ourselves to estimators $M$ which output in $V = \{\Sigma^{-\frac{1}{2}} v | v \in [-\delta_*\sigma, \delta_*\sigma]^{\sqrt{d}} \times \{0\}^{d-\sqrt{d}}\}$ almost surely – otherwise the error can be reduced by projected to this region. Thus ess-sup$\|M(\boldsymbol{Z}) - \beta_P\|_\Sigma^2 \leq 4\sqrt{d}\delta_*^2\sigma^2$ for any estimator $M$ and any distribution $P$. Thus, combining the above observations, we have

$$\inf_M \sup_{P \in \mathcal{P}_{\delta_*}} \mathbb{E}_{\boldsymbol{Z} \sim_\lambda P} \|M(\boldsymbol{Z}) - \mu_P\|_\Sigma^2 \leq \inf_M \sup_{P \in \mathcal{P}_{\delta_*}} \frac{4}{5} Q\left( \|M(\boldsymbol{Z}) - \mu_P\|_\Sigma^2, \frac{4}{5} \right) + \frac{4\sqrt{d}\delta_*^2\sigma^2}{5}. \tag{172}$$

Using (171), we get

$$\frac{7}{8} \sqrt{d}\delta_*^2\sigma^2 \leq \frac{4}{5} \inf_M \sup_{P \in \mathcal{P}_{\sigma\delta_*}} Q\left( \|M(\boldsymbol{Z}) - \mu_P\|_\Sigma^2, \frac{4}{5} \right) + \frac{4\sqrt{d}\delta_*^2\sigma^2}{5} \tag{173}$$

$$= \frac{4}{5} L_{\text{reg}}(\boldsymbol{\lambda}, \mathcal{P}_{\sigma\delta_*}) + \frac{4\sqrt{d}\delta_*^2\sigma^2}{5} \tag{174}$$

$$\implies \frac{3}{40} \sqrt{d}\delta_*^2\sigma^2 \leq L_{\text{reg}}(\boldsymbol{\lambda}, \mathcal{P}_{\sigma\delta_*}) \leq L_{\text{reg}}(\boldsymbol{\lambda}, \mathcal{D}(\Sigma, \sigma^2)). \tag{175}$$

### D.3  Minimax Optimality

For proving Theorem 5, we shall use the weighing $w_i = \frac{\mathbb{I}\{\lambda_i \leq 1 - e^{-\frac{2\delta_* \sqrt{d}}{\pi}}\}}{N(1 - e^{-\frac{2\delta_* \sqrt{d}}{\pi}})}$ in our upper bound.

From (137), $\forall t \in [0, 1]$ such that $t^2 + \frac{d}{N(t)} \leq c$ for some universal constant $c$, we have

$$\|\hat{\beta}_{\mathsf{S}}(\boldsymbol{Z}, w(t)) - \mu\|_{\hat{\Sigma}}^2 \lesssim \sigma^2 \left( t^2 + \frac{d}{N(t)} \right). \tag{176}$$

Thus, if $\min_{t \in [0,1]} \left( t^2 + \frac{d}{N(t)} \right) \leq c$ and let the minimum value be attained at $t^*$, then we have

$$\|\hat{\beta}_{\mathsf{S}}(\boldsymbol{Z}, w(t^*)) - \mu\|_{\hat{\Sigma}}^2 \lesssim \sigma^2 \left( t^{*2} + \frac{d}{N(t^*)} \right) \tag{177}$$

$$\leq \sigma^2 \left( 1 - e^{-\frac{2\delta_* \sqrt{d}}{\pi}} \right)^2 + \sigma^2 \frac{d}{N \left( 1 - e^{-\frac{2\delta_* \sqrt{d}}{\pi}} \right)} \tag{178}$$

$$\leq \sigma^2 \frac{4d\delta_*^2}{\pi^2} + 64\sigma^2 d\delta_*^2 \tag{179}$$

$$\simeq \sigma^2 d\delta_*^2, \tag{180}$$

where we used (170) in (179). Combining with (175), when $\min_{t \in [0,1]} \left( t^2 + \frac{d}{N(t)} \right) \leq c$, we have

$$\sqrt{d}\sigma^2 \delta_*^2 \lesssim L_{\mathsf{reg}}(\boldsymbol{\lambda}, \mathcal{D}(\Sigma, \sigma^2)) \lesssim \sigma^2 \min_{t \in [0,1]} \left( t^2 + \frac{d}{N(t)} \right) \leq d\sigma^2 \delta_*^2. \tag{181}$$

Thus, when $\min_{t \in [0,1]} \left( t^2 + \frac{d}{N(t)} \right) \leq c$,

$$\frac{\sigma^2}{\sqrt{d}} \min_{t \in [0,1]} \left( t^2 + \frac{d}{N(t)} \right) \lesssim L_{\mathsf{reg}}(\boldsymbol{\lambda}, \mathcal{D}(\Sigma, \sigma^2)) \lesssim \sigma^2 \min_{t \in [0,1]} \left( t^2 + \frac{d}{N(t)} \right). \tag{182}$$

## E  Weighted Generalization Bound

We refer the readers to [43] for an introduction to VC dimension, covering numbers, and packing numbers.

Define the weighted empirical Rademacher complexity to be $\mathcal{R}_w(\mathcal{F} \circ \boldsymbol{X}) = \mathbb{E}\left[\sup_{f \in \mathcal{F}} \sum_i w_i \sigma_i f(X_i) \big| \boldsymbol{X}\right]$. We first generalize Massart's lemma to weighted Rademacher complexity.

**Lemma 1** (Weighted Massart's Lemma)**.** *Assume $|\mathcal{F}|$ is finite and let*

$$B_w = \max_{f \in \mathcal{F}} \sqrt{\sum_{i=1}^{n} w_i^2 f(X_i)}, \tag{183}$$

*then*

$$\mathcal{R}_w(\mathcal{F} \circ \boldsymbol{X}) \leq B_w \sqrt{2 \ln |\mathcal{F}|} \tag{184}$$

*Proof.*

$$e^{s\mathcal{R}_w(\mathcal{F}\circ\mathbf{X})} = e^{s\mathbb{E}\left[\sup_{f\in\mathcal{F}}\sum_i w_i\sigma_i f(X_i)\big|\mathbf{X}\right]} \tag{185}$$

$$\leq \mathbb{E}\left[e^{s\sup_{f\in\mathcal{F}}\sum_i w_i\sigma_i f(X_i)}\big|\mathbf{X}\right] \quad \text{(conditional Jensen's inequality)} \tag{186}$$

$$\leq \mathbb{E}\left[\sum_{f\in\mathcal{F}} e^{s\sum_i w_i\sigma_i f(X_i)}\big|\mathbf{X}\right] \tag{187}$$

$$= \sum_{f\in\mathcal{F}}\prod_{i=1}^n \mathbb{E}\left[e^{sw_i\sigma_i f(X_i)}\big|\mathbf{X}\right] \tag{188}$$

$$\leq \sum_{f\in\mathcal{F}}\prod_{i=1}^n e^{s^2 w_i^2 f(X_i)^2/2} \quad \text{(by Hoeffding's lemma)} \tag{189}$$

$$\leq |\mathcal{F}|e^{s^2 B_w^2/2}. \tag{190}$$

Thus,

$$\mathcal{R}_w(\mathcal{F}\circ\mathbf{X}) \leq \frac{\ln|\mathcal{F}|}{s} + \frac{sB_w^2}{2} \ \forall s > 0. \tag{191}$$

Setting $s = \frac{\sqrt{2\ln|\mathcal{F}|}}{B_w}$, we get the claimed upper bound. $\qquad\square$

**Lemma 2** (Weighted Rademacher Complexity Bound). $\mathcal{R}_w(\mathcal{F}\circ\mathbf{X}) \leq 31\|w\|_2\sqrt{\mathsf{VC}(\mathcal{F})} \ \forall\mathbf{X}$, *where* $\mathsf{VC}(\mathcal{F})$ *is the Vapnik–Chervonenkis dimension of* $\mathcal{F}$.

*Proof.* Much of this proof follows the analysis of Liao [44], Rebeschini [45], adapted to the weighted version of our problem.

In the context of this proof, for the realized $\mathbf{X}$ and for $f \in \mathcal{F}$, define

$$\|f\|_w = \frac{\sqrt{\sum_{i=1}^n w_i^2 f(X_i)^2}}{\|w\|_2}. \tag{192}$$

Let $\max_{f\in\mathcal{F}}\|f\|_w \leq c$. For family of functions that we consider, $c = 1$.

Let $\mathcal{F}_j \subseteq \mathcal{F}$ be the minimal $\epsilon_j = \frac{c}{2^j}$ cover of $\mathcal{F}$ with respect to $\|\cdot\|_w$, i.e., $\mathcal{F}_j$ is the set of least cardinality such that for any $f \in \mathcal{F}$, $\exists f' \in \mathcal{F}_j$ such that $\|f - f'\|_w \leq \epsilon_j$. Let $|\mathcal{F}_j| = C(\mathcal{F}, \epsilon_j, \|\cdot\|_w)$.

For any $f \in \mathcal{F}$, let $f_j(f) \in \mathcal{F}_j$ be such that $\|f - f_j(f)\|_w \leq \epsilon_j$. Denoting $\mathbb{E}[\cdot|\mathbf{X}]$ as $\mathbb{E}_{\boldsymbol{\sigma}}[\cdot]$, we have for any $m \geq 1$,

$$\mathcal{R}_w(\mathcal{F}\circ\mathbf{X}) = \mathbb{E}_{\boldsymbol{\sigma}}\left[\sup_{f\in\mathcal{F}}\sum_i w_i\sigma_i\left(f(X_i) - f_m(f)(X_i) + \sum_{j=1}^m (f_j(f)(X_i) - f_{j-1}(f)(X_i))\right)\right] \tag{193}$$

$$\leq \mathbb{E}_{\boldsymbol{\sigma}}\left[\sup_{f\in\mathcal{F}}\sum_i w_i\sigma_i(f(X_i) - f_m(f)(X_i))\right] + \mathbb{E}_{\boldsymbol{\sigma}}\left[\sup_{f\in\mathcal{F}}\sum_i w_i\sigma_i\sum_{j=1}^m f_j(f)(X_i) - f_{j-1}(f)(X_i)\right]. \tag{194}$$

Notice that

$$\mathbb{E}_{\boldsymbol{\sigma}}\left[\sup_{f\in\mathcal{F}}\sum_i w_i\sigma_i(f(X_i) - f_m(f)(X_i))\right] \leq \sup_{f\in\mathcal{F}}\sum_i w_i|f(X_i) - f_m(f)(X_i)| \tag{195}$$

$$\leq \sup_{f\in\mathcal{F}}\sqrt{n}\|w\|_2\|f - f_m(f)\|_w \tag{196}$$

$$\leq \sqrt{n}\|w\|_2\epsilon_m. \tag{197}$$

Now, for the second term, by Lemma 1, we have

$$\sup_{f \in \mathcal{F}} \sum_i w_i \sigma_i \sum_{j=1}^m f_j(f)(X_i) - f_{j-1}(f)(X_i) \le \sum_{j=1}^m \sup_{f \in \mathcal{F}} \sum_i w_i \sigma_i \left( f_j(f)(X_i) - f_{j-1}(f)(X_i) \right). \tag{198}$$

Thus, using Lemma 1,

$$\mathbb{E}_{\boldsymbol{\sigma}} \left[ \sup_{f \in \mathcal{F}} \sum_i w_i \sigma_i \sum_{j=1}^m f_j(f)(X_i) - f_{j-1}(f)(X_i) \right] \le \tag{199}$$

$$\sum_{j=1}^m \|w\|_2 \sup_{f \in \mathcal{F}} \|f_j(f) - f_{j-1}(f)\|_w \sqrt{2 \ln |\{f_j(f) - f_{j-1}(f) | f \in \mathcal{F}\}|}. \tag{200}$$

Note that $|\{f_j(f) - f_{j-1}(f) | f \in \mathcal{F}\}| \le |\mathcal{F}_j||\mathcal{F}_{j-1}| \le 2 \ln |\mathcal{F}_j|$. Further, by triangle inequality,

$$\|f_j(f) - f_{j-1}(f)\|_w \le \|f_j(f) - f\|_w + \|f - f_{j-1}(f)\|_w \tag{201}$$

$$\le \epsilon_j + \epsilon_{j-1} \tag{202}$$

$$= 3\epsilon_j \tag{203}$$

$$= 6(\epsilon_j - \epsilon_{j+1}). \tag{204}$$

Thus,

$$\mathbb{E}_{\boldsymbol{\sigma}} \left[ \sup_{f \in \mathcal{F}} \sum_i w_i \sigma_i \sum_{j=1}^m f_j(f)(X_i) - f_{j-1}(f)(X_i) \right] \le 12\|w\|_2 \sum_{j=1}^m (\epsilon_j - \epsilon_{j+1}) \sqrt{\ln |\mathcal{F}_j|}. \tag{205}$$

Combining the above bounds,

$$\mathcal{R}_w(\mathcal{F} \circ \boldsymbol{X}) \le \|w\|_2 \left\{ 2\sqrt{n}\epsilon_{m+1} + 12 \sum_{j=1}^m (\epsilon_j - \epsilon_{j+1}) \sqrt{\ln C(\mathcal{F}, \epsilon_j, \|\cdot\|_w)} \right\} \tag{206}$$

$$\le \|w\|_2 \left\{ 2\sqrt{n} \frac{c}{2^{m+1}} + 12 \int_{c/2^{m+1}}^{c/2} \sqrt{\ln C(\mathcal{F}, x, \|\cdot\|_w)} dx \right\}. \tag{207}$$

For any $\epsilon \in (0, \frac{c}{2}]$, $\exists m \in \mathbb{Z}_{\ge 1}$ such that $\frac{c}{2^{m+1}} \le \epsilon \le \frac{c}{2^m}$. Thus,

$$\mathcal{R}_w(\mathcal{F} \circ \boldsymbol{X}) \le \|w\|_2 \left\{ 2\sqrt{n}\epsilon + 12 \int_{\epsilon/2}^{c/2} \sqrt{\ln C(\mathcal{F}, x, \|\cdot\|_w)} dx \right\}. \tag{208}$$

Taking $\epsilon \to 0$, obtain

$$\mathcal{R}_w(\mathcal{F} \circ \boldsymbol{X}) \le 12\|w\|_2 \int_0^{c/2} \sqrt{\ln C(\mathcal{F}, x, \|\cdot\|_w)} dx. \tag{209}$$

Setting $c = 1$ and using Lemma 3 obtain

$$\mathcal{R}_w(\mathcal{F} \circ \boldsymbol{X}) \le 12\sqrt{\mathsf{VC}(\mathcal{F})}\|w\|_2 \int_0^{c/2} \sqrt{\frac{10}{x^2} \ln \frac{2e}{x^2}} dx \tag{210}$$

$$\le 12\sqrt{\mathsf{VC}(\mathcal{F})}\|w\|_2 \int_0^{c/2} \sqrt{\ln \frac{20}{x^4}} dx \quad \text{using } \ln x \le x/e \, \forall x \ge e \tag{211}$$

$$\le 31\sqrt{\mathsf{VC}(\mathcal{F})}\|w\|_2. \tag{212}$$

$$\square$$

**Lemma 3.** $C(\mathcal{F}, x, \|\cdot\|_w) \le \left[\frac{10}{x^2} \ln \frac{2e}{x^2}\right]^{\mathsf{VC}(\mathcal{F})}$

The readers can refer to the proof of Lemma 3 presented in [43, 45] which generalizes to our modified distance $\|\cdot\|_w$.

**Proposition 4.** *Let $\mathcal{F}$ be a family of $0-1$ valued functions and let $X_1, \ldots, X_n \overset{iid}{\sim} P$. For a fixed vector $w \in \mathbb{R}^d$, with probability at least $1-\delta$,*

$$\sup_{f \in \mathcal{F}} \left\{ \sum_i w_i(f(X_i) - \mathbb{E}[f(X)]) \right\} \leq 62\|w\|_2\sqrt{\mathsf{VC}(\mathcal{F})} + \|w\|_2\sqrt{\frac{\log\frac{1}{\delta}}{2}}, \qquad (213)$$

*Proof.* By McDiarmid's inequality, with probability at least $1-\delta$,

$$\sup_{f \in \mathcal{F}} \left\{ \sum_i w_i(f(X_i) - \mathbb{E}[f(X)]) \right\} \leq \mathbb{E}\left[\sup_{f \in \mathcal{F}} \left\{ \sum_i w_i(f(X_i) - \mathbb{E}[f(X)]) \right\}\right] + \|w\|_2\sqrt{\frac{\log\frac{1}{\delta}}{2}}. \tag{214}$$

Let $X_1', \ldots, X_n' \overset{iid}{\sim} P$ then

$$\mathbb{E}\left[\sup_{f \in \mathcal{F}} \left\{ \sum_i w_i(f(X_i) - \mathbb{E}[f(X)]) \right\}\right] = \mathbb{E}\left[\sup_{f \in \mathcal{F}} \left\{ \mathbb{E}\left[ \sum_i w_i(f(X_i) - f(X_i')) \Big| \boldsymbol{X} \right] \right\}\right] \tag{215}$$

$$\leq \mathbb{E}\left[\sup_{f \in \mathcal{F}} \left\{ \sum_i w_i(f(X_i) - f(X_i')) \right\}\right] \tag{216}$$

by Jensen's inequality. Let $\sigma_1, \ldots, \sigma_n$ be i.i.d. Rademacher random variables then

$$\mathbb{E}\left[\sup_{f \in \mathcal{F}} \left\{ \sum_i w_i(f(X_i) - f(X_i')) \right\}\right] = \mathbb{E}\left[\sup_{f \in \mathcal{F}} \left\{ \sum_i w_i\sigma_i(f(X_i) - f(X_i')) \right\}\right] \tag{217}$$

$$\leq \mathbb{E}\left[\sup_{f \in \mathcal{F}} \sum_i w_i\sigma_i f(X_i) + \sup_{f \in \mathcal{F}} \sum_i w_i(-\sigma_i)f(X_i')\right] \tag{218}$$

$$= 2\mathbb{E}\left[\sup_{f \in \mathcal{F}} \sum_i w_i\sigma_i f(X_i)\right] \tag{219}$$

$$= 2\mathbb{E}\left[\mathcal{R}_w(\mathcal{F} \circ \boldsymbol{X})\right]. \tag{220}$$

The result of Lemma 2 completes the proof. $\qquad\square$

# F  LeCam-Assouad Interpolation Lower Bound

In this section, we provide an improved lower bound over those in Appendix C.2 and Appendix D.2, while using similar ideas.

Fix a value of $t \in [d]$ and $m = \frac{d}{t}$ such that they are whole numbers. Roughly speaking, we shall parametrize the mean of the $d$-dimensional Gaussian distributions in the hypotheses by vectors in $\{-\delta, \delta\}^t$ with repeating each element $m$ times. Choosing $t = 1$ leads to Le Cam's method while $t = d$ leads to Assouad's method.

For a vector $v \in \mathbb{R}^t$, let $e_t(v) \in \mathbb{R}^d$ with

$$e_t(v)_i = v_{\lceil i/t \rceil} \ \ \forall i \in [d], \tag{221}$$

where $\lceil \cdot \rceil$ denotes the ceiling function. For convenience, we drop the subscript and denote $e_t(v)$ by $e(v)$. Define the parameterized distribution $P_\delta(\tau) = \mathcal{N}(\delta e(\tau), I)$ for $\delta > 0$ to be specified later, and let $\mathcal{P}_\delta = \{P_\delta(\tau) | \tau \in \{-1, 1\}^t\}$.

**Note:** The supremum in our minimax definition is over both the true distribution and the adversarial strategy. We shall specify the adversarial strategy later for our lower bound argument.

Note that $L_{\mathsf{PAC}}(\boldsymbol{\lambda}, \mathcal{D}_d^{\mathcal{N}}) \geq L_{\mathsf{PAC}}(\boldsymbol{\lambda}, \mathcal{P}_\delta) \ \forall \delta$ and thus, we shall lower bound $L_{\mathsf{PAC}}(\boldsymbol{\lambda}, \mathcal{P}_\delta)$.

We begin by lower bounding

$$L_{\mathbb{E}}(\boldsymbol{\lambda}, \mathcal{P}_\delta) = \inf_M \sup_{P \in \mathcal{P}_\delta} \mathbb{E}_{\boldsymbol{Z} \sim_\lambda P}\left[\|M(\boldsymbol{Z}) - \mu_P\|_2^2\right]. \tag{222}$$

For a vector $\tau$, let $\tau'^j$ denote the vector such that $\tau_i = \tau_i'^j \forall i \neq j$ and $\tau_j = -\tau_j'^j$. Using Assouad's lower bound technique [28],

$$\inf_M \sup_{P \in \mathcal{P}_\delta} \mathbb{E}_{\mathbf{Z} \sim_\lambda P} \left[ \|M(\mathbf{Z}) - \mu_P\|_2^2 \right] \geq \inf_M \frac{1}{2^t} \sum_{\tau \in \{-1,1\}^t} \mathbb{E} \left[ \|M(\mathbf{Z}) - \delta e(\tau)\|_2^2 \right] \tag{223}$$

$$= \inf_M \frac{1}{2^t} \sum_{\tau \in \{-1,1\}^t} \sum_{j=1}^d \mathbb{E} \left[ |M(\mathbf{Z})_j - \delta e(\tau)_j|^2 \right] \tag{224}$$

$$= \inf_M \sum_{k=1}^m \sum_{i=1}^t \frac{1}{2^t} \sum_{\tau \in \{-1,1\}^t} \mathbb{E} \left[ |M(\mathbf{Z})_{(i-1)t+k} - \delta \tau_i|^2 \right] \tag{225}$$

$$\geq \sum_{k=1}^m \inf_M \sum_{i=1}^t \frac{1}{2^t} \sum_{\tau \in \{-1,1\}^t} \mathbb{E} \left[ |M(\mathbf{Z})_{(i-1)t+k} - \delta \tau_i|^2 \right] \tag{226}$$

Now, note that $\forall i, k$

$$\sum_{\tau \in \{-1,1\}^t} \mathbb{E} \left[ |M(\mathbf{Z})_{(i-1)t+k} - \delta \tau_i|^2 \right] = \sum_{\tau \in \{-1,1\}^t} \frac{\mathbb{E} \left[ |M(\mathbf{Z})_{(i-1)t+k} - \delta \tau_i|^2 \right] + \mathbb{E} \left[ |M(\mathbf{Z})_{(i-1)t+k} - \delta \tau_i'^i|^2 \right]}{2} \tag{227}$$

Thus, we get

$$L_{\mathbb{E}}(\lambda, \mathcal{P}_\delta) \geq \sum_{k=1}^m \inf_M \sum_{i=1}^t \frac{1}{2^t} \sum_{\tau \in \{-1,1\}^t} \frac{\mathbb{E} \left[ |M(\mathbf{Z})_{(i-1)t+k} - \delta \tau_i|^2 \right] + \mathbb{E} \left[ |M(\mathbf{Z})_{(i-1)t+k} - \delta \tau_i'^i|^2 \right]}{2} \tag{228}$$

$$\geq \sum_{k=1}^m \sum_{i=1}^t \frac{1}{2^t} \sum_{\tau \in \{-1,1\}^t} \inf_M \frac{\mathbb{E} \left[ |M(\mathbf{Z})_{(i-1)t+k} - \delta \tau_i|^2 \right] + \mathbb{E} \left[ |M(\mathbf{Z})_{(i-1)t+k} - \delta \tau_i'^i|^2 \right]}{2} \tag{229}$$

$$\geq \sum_{k=1}^m \sum_{i=1}^t \frac{1}{2^t} \sum_{\tau \in \{-1,1\}^t} \delta^2 (1 - \mathsf{TV}(P_{\mathbf{Z} \sim_\lambda P_\delta(\tau)}, P_{\mathbf{Z} \sim_\lambda P_\delta(\tau'^i)})), \tag{230}$$

where the last line follows by Le Cam's method and the notation $P_{\mathbf{Z} \sim_\lambda P_\delta(\tau)}$ refers to the distribution of $\mathbf{Z}$ when the true distribution is $P_\delta(\tau)$.

**Adversarial strategy motivation:** Consider a particular sample with corruption rate $\lambda$ and let the underlying true distribution be $P_\delta(\tau)$. Denote the perturbed sample as $Z(\tau)$ and the outlier to be $\tilde{X}(\tau)$. Note that $Z(\tau) \sim (1 - \lambda) P_{X(\tau)} + \lambda P_{\tilde{X}(\tau)}$. For any particular clean sample $X(\tau) \sim P_\delta(\tau)$, the adversary's goal is to ensure the sample $Z(\tau)$ contains no information about $\tau$. One possible way is that the adversary can try to ensure $Z(\tau) \sim \frac{\max_{\tau'} P(\tau')}{T}$, where the maximum is taken pointwise over the pdf and $T$ is a normalizing constant. Observe the identity

$$P(\tau) + \left( \max_{\tau' \neq \tau} P(\tau') - P(\tau) \right)_+ = \max_{\tau'} P(\tau'), \tag{231}$$

where $(\cdot)_+ := \max\{\cdot, 0\}$ is pointwise over the pdf. First, note that

$$\int_{x \in \mathbb{R}^d} \left( \max_{\tau' \neq \tau} P_\delta(\tau')(x) - P_\delta(\tau)(x) \right)_+ dx = T - 1,$$

where $P_\delta(\tau)(x)$ is understood to be the pdf of $P_\delta(\tau)$ at $x$. Thus, $\frac{\left( \max_{\tau' \neq \tau} P(\tau') - P(\tau) \right)_+}{T-1}$ is a valid pdf, and for $\lambda = \frac{T-1}{T}$, we have $(1 - \lambda) P(\tau) + \lambda \frac{\left( \max_{\tau' \neq \tau} P(\tau') - P(\tau) \right)_+}{T-1} = \frac{\max_{\tau'} P(\tau')}{T}$. Thus, for

any $\lambda \geq 1 - \frac{1}{T}$, there exists a way for the adversary to make the distribution of $Z(\tau) \sim \frac{\max_{\tau'} P(\tau')}{T}$, rendering the sample useless for identifying $\tau$.

Now, we find an upper bound on $T$ in terms of $\delta$. Let $\mathbf{1}$ denote a vector of size $m$ with all elements equal to 1.

$$T = \int_{x \in \mathbb{R}^d} \max_{\tau \in \{-1,1\}^t} \frac{1}{\sqrt{(2\pi)^d}} e^{-\frac{\|x - \delta e(\tau)\|_2^2}{2}} dx \tag{232}$$

$$= \int_{x \in \mathbb{R}^d} \frac{1}{\sqrt{(2\pi)^d}} \max_{\tau \in \{-1,1\}^t} e^{-\frac{\|x - \delta e(\tau)\|_2^2}{2}} dx \tag{233}$$

$$= \left[ \int_{x' \in \mathbb{R}^m} \frac{1}{\sqrt{(2\pi)^m}} \max_{\tau \in \{-1,1\}} e^{-\frac{\|x' - \delta \tau \mathbf{1}\|_2^2}{2}} dx' \right]^t . \tag{234}$$

Now note that

$$\int_{x' \in \mathbb{R}^m} \frac{1}{\sqrt{(2\pi)^m}} \max_{\tau \in \{-1,1\}} e^{-\frac{\|x' - \delta \tau \mathbf{1}\|_2^2}{2}} dx' = 1 + \mathsf{TV}(\mathcal{N}(-\delta\mathbf{1}, I), \mathcal{N}(\delta\mathbf{1}, I)), \tag{235}$$

by total variation definition. Using Pinsker's inequality and KL divergence between multivariate Gaussians, we obtain

$$\mathsf{TV}(\mathcal{N}(-\delta\mathbf{1}, I), \mathcal{N}(\delta\mathbf{1}, I)) \leq \sqrt{\frac{\mathsf{KL}(\mathcal{N}(-\delta\mathbf{1}, I), \mathcal{N}(\delta\mathbf{1}, I))}{2}} \tag{236}$$

$$= \delta\sqrt{m} \tag{237}$$

Combining (234), (235) and (237), and using $m = \frac{d}{t}$

$$T \leq (1 + \delta\sqrt{m})^t \tag{238}$$

$$\leq e^{\delta\sqrt{td}}. \tag{239}$$

Thus, if $\lambda \geq 1 - e^{-\delta\sqrt{td}}$ then the adversary can ensure that $Z(\tau) \sim \frac{\max_{\tau'} P(\tau')}{T}$ and contains no information about $\tau$.

**Adversarial strategy:** If for sample $i$, $\lambda_i \geq 1 - e^{-\delta\sqrt{td}}$, then independently sample $\tilde{X}(\tau) \sim \beta \left( \frac{\max_{\tau' \neq \tau} P_\delta(\tau') - P_\delta(\tau)}{T - 1} \right)_+ + (1 - \beta)\frac{\max_{\tau'} P_\delta(\tau')}{T}$, where $\beta = \frac{(T-1)(1-\lambda)}{\lambda}$. The constraint $\lambda_i \geq 1 - e^{-\delta\sqrt{td}}$ ensures $\beta \in [0, 1]$ and the above is a valid distribution.

Thus, $Z(\tau) \sim \frac{\max_{\tau'} P_\delta(\tau')}{T}$. If $\lambda_i < 1 - e^{-\delta\sqrt{td}}$, then adversary does no corruption, or equivalently, independently sample $\tilde{X}(\tau) \sim P_\delta(\tau)$ so that $Z(\tau) \sim P_\delta(\tau)$.

Thus, using Pinsker's inequality in (230), obtain

$$L_{\mathbb{E}}(\boldsymbol{\lambda}, \mathcal{P}_\delta) \geq \sum_{k=1}^{m} \sum_{i=1}^{t} \frac{1}{2^t} \sum_{\tau \in \{-1,1\}^t} \delta^2 \left( 1 - \sqrt{\frac{1}{2}\mathsf{KL}(P_{\mathbf{Z} \sim \lambda P_\delta(\tau)}, P_{\mathbf{Z} \sim \lambda P_\delta(\tau'^i)})} \right) \tag{240}$$

$$= \sum_{k=1}^{m} \sum_{i=1}^{t} \frac{1}{2^t} \sum_{\tau \in \{-1,1\}^t} \delta^2 \left( 1 - \sqrt{\frac{1}{2}\mathsf{KL}\left( \otimes_{j=1}^{n} P_{Z_j(\tau)}, \otimes_{j=1}^{n} P_{Z_i(\tau'^i)} \right)} \right) \tag{241}$$

$$= \sum_{k=1}^{m} \sum_{i=1}^{t} \frac{1}{2^t} \sum_{\tau \in \{-1,1\}^t} \delta^2 \left( 1 - \sqrt{\frac{1}{2} \sum_{j:\lambda_j < 1 - e^{-\delta\sqrt{dt}}} \mathsf{KL}\left( P_\delta(\tau), P_\delta(\tau'^i) \right)} \right) \tag{242}$$

$$= d\delta^2 \left( 1 - \sqrt{\frac{d\delta^2}{t} n(1 - e^{-\delta\sqrt{dt}})} \right) \quad \forall \delta \tag{243}$$

Let $\delta_*$ be such that

$$n(1 - e^{-\delta_* \sqrt{dt}}) \leq \frac{t}{64 d \delta_*^2}, \quad \text{and } N(1 - e^{-\delta_* \sqrt{dt}}) \geq \frac{t}{64 d \delta_*^2}. \tag{244}$$

Substituting $\delta_*$ in (243),

$$\frac{7}{8}d\delta_*^2 \leq \inf_M \sup_{P \in \mathcal{P}_{\delta_*}} \mathbb{E}_{\boldsymbol{Z} \sim_\lambda P}\left[\|M(\boldsymbol{Z}) - \mu_P\|_2^2\right]. \tag{245}$$

For a random variable $K$, let $Q(K, \alpha) := \inf\{t : \Pr\left[K \geq t\right] \leq (1-\alpha)\}$, i.e., $Q(\cdot, \alpha)$ is the $\alpha$-quantile of the random variable. Note that for a random variable $K$, $\mathbb{E}K \leq Q(K, x)x + (1 - x)\text{ess-sup}K$ $\forall x \in [0, 1]$, where ess-sup denotes essential supremum. Further, note that in the minimax term $\inf_M \sup_{P \in \mathcal{P}_{\delta_*}}$, we can restrict ourselves to estimators $M$ which output in $[-\delta_*, \delta_*]^d$ almost surely – otherwise the error can be reduced by projected to this region. Thus ess-sup$\|M(\boldsymbol{Z}) - \mu_P\|_2^2 \leq 4d\delta_*^2$ for any estimator $M$ and any distribution $P$. Thus, combining the above observations, we have

$$\inf_M \sup_{P \in \mathcal{P}_{\delta_*}} \mathbb{E}_{\boldsymbol{Z} \sim_\lambda P}\|M(\boldsymbol{Z}) - \mu_P\|_2^2 \leq \inf_M \sup_{P \in \mathcal{P}_{\delta_*}} \frac{4}{5}Q\left(\|M(\boldsymbol{Z}) - \mu_P\|_2^2, \frac{4}{5}\right) + \frac{4d\delta_*^2}{5}. \tag{246}$$

Using (245), we get

$$\frac{7}{8}d\delta_*^2 \leq \frac{4}{5}\inf_M \sup_{P \in \mathcal{P}_{\delta_*}} Q\left(\|M(\boldsymbol{Z}) - \mu_P\|_2^2, \frac{4}{5}\right) + \frac{4d\delta_*^2}{5} \tag{247}$$

$$= \frac{4}{5}L_{\text{PAC}}(\boldsymbol{\lambda}, \mathcal{P}_{\delta_*}) + \frac{4d\delta_*^2}{5} \tag{248}$$

$$\implies \frac{3}{40}d\delta_*^2 \leq L_{\text{PAC}}(\boldsymbol{\lambda}, \mathcal{P}_{\delta_*}) \leq L_{\text{PAC}}(\boldsymbol{\lambda}, \mathcal{D}_d^{\mathcal{N}}). \tag{249}$$

We remark that the lower bound presented here appears to be different from that of Appendix C.2. However, the exact difference is hard to comment on since the term $\delta_*$ is implicitly defined in the two bounds. We do however remark that we can reconstruct the same $\sqrt{d}$ gap in upper and lower bound using this method as demonstrated below.

## F.1   On Minimax Optimality

In the upper bound, use the weighing $w_i = \frac{\mathbb{I}\{\lambda_i \leq 1 - e^{\delta_* \sqrt{dt}}\}}{N(1 - e^{\delta_* \sqrt{dt}})}$. From (74), $\forall v \in [0, 1]$ such that $v^2 + \frac{d}{N(v)} \leq c$ for some universal constant $c$, we have

$$\|\hat{\mu}_{\text{S}}(\boldsymbol{Z}, w(v)) - \mu\|_2^2 \lesssim \left(v^2 + \frac{d}{N(v)}\right). \tag{250}$$

Thus, if $\min_{v \in [0,1]}\left(v^2 + \frac{d}{N(v)}\right) \leq c$ and let the minimum value be attained at $v^*$, then we have

$$\|\hat{\mu}_{\text{S}}(\boldsymbol{Z}, w(v^*)) - \mu\|_2^2 \lesssim \left(v^{*2} + \frac{d}{N(v^*)}\right) \tag{251}$$

$$\leq \left(1 - e^{-\delta_* \sqrt{dt}}\right)^2 + \frac{d}{N\left(1 - e^{-\delta_* \sqrt{dt}}\right)} \tag{252}$$

$$\leq dt\delta_*^2 + \frac{64d^2\delta_*^2}{t} \tag{253}$$

$$\simeq d^{1.5}\delta_*^2 \quad (\text{setting } t = \sqrt{d}) \tag{254}$$

where we used (244) in (253). Combining with (249), when $\min_{v \in [0,1]}\left(v^2 + \frac{d}{N(v)}\right) \leq c$, we have

$$d\delta_*^2 \lesssim L_{\text{PAC}}(\boldsymbol{\lambda}, \mathcal{D}_d^{\mathcal{N}}) \lesssim \min_{v \in [0,1]}\left(v^2 + \frac{d}{N(v)}\right) \leq d^{1.5}\delta_*^2. \tag{255}$$

Thus, when $\min_{t \in [0,1]}\left(t^2 + \frac{d}{N(t)}\right) \leq c$,

$$\frac{1}{\sqrt{d}}\min_{t \in [0,1]}\left(t^2 + \frac{d}{N(t)}\right) \lesssim L_{\text{PAC}}(\boldsymbol{\lambda}, \mathcal{D}_d^{\mathcal{N}}) \lesssim \min_{t \in [0,1]}\left(t^2 + \frac{d}{N(t)}\right). \tag{256}$$

## F.2 Benefits of this Lower Bound

The lower bound presented in Appendix C.2 does not match the standard lower bound when all the corruptions rates are equal, i.e., the homogeneous robust estimation problem. When $\boldsymbol{\lambda} = \lambda\mathbf{1}$, from (95) in Appendix C.2, we obtain

$$L_{\mathbb{E}}(\lambda\mathbf{1}, \mathcal{P}_\delta) \geq \sqrt{d}\delta^2 \left(1 - \sqrt{n\delta^2 \mathbb{I}\left\{\sqrt{d}\delta < \log\left(\frac{1}{1-\lambda}\right)\right\}}\right). \tag{257}$$

Setting $\delta_* = \max\left\{\frac{1}{\sqrt{2n}}, \frac{1}{\sqrt{d}}\log\left(\frac{1}{1-\lambda}\right)\right\}$, obtain the lower bound

$$L_{\mathbb{E}}(\lambda\mathbf{1}, \mathcal{P}_{\delta_*}) \gtrsim \frac{\sqrt{d}}{n} + \frac{\lambda^2}{\sqrt{d}}, \tag{258}$$

and thus,

$$L_{\mathsf{PAC}}(\lambda\mathbf{1}, \mathcal{D}_d^{\mathcal{N}}) \gtrsim \frac{\sqrt{d}}{n} + \frac{\lambda^2}{\sqrt{d}}. \tag{259}$$

Using the lower bound presented of (243), we can fix this issue. Take $t = 1$ and set $\delta_* = \max\left\{\frac{1}{\sqrt{2dn}}, \frac{1}{\sqrt{d}}\log\left(\frac{1}{1-\lambda}\right)\right\}$ to get the lower bound

$$L_{\mathbb{E}}(\lambda\mathbf{1}, \mathcal{P}_{\delta_*}) \gtrsim \frac{1}{n} + \lambda^2. \tag{260}$$

Now take $t = d$ and set $\delta_* = \max\left\{\frac{1}{\sqrt{2n}}, \frac{1}{d}\log\left(\frac{1}{1-\lambda}\right)\right\}$ to get the lower bound

$$L_{\mathbb{E}}(\lambda\mathbf{1}, \mathcal{P}_{\delta_*}) \gtrsim \frac{d}{n} + \frac{\lambda^2}{d}. \tag{261}$$

Combining the two cases, obtain the bound

$$L_{\mathbb{E}}(\lambda\mathbf{1}, \mathcal{P}_{\delta_*}) \gtrsim \frac{d}{n} + \lambda^2, \tag{262}$$

and thus, we get the right lower bound of

$$L_{\mathsf{PAC}}(\lambda\mathbf{1}, \mathcal{D}_d^{\mathcal{N}}) \gtrsim \frac{d}{n} + \lambda^2. \tag{263}$$

Similar steps can be used for Gaussian linear regression.

