# OpenReview forum: "Robust Estimation Under Heterogeneous Corruption Rates"
_NeurIPS.cc/2025/Conference — NeurIPS 2025 poster_

### Official Review · Reviewer_9oan · 2025-06-28

**Clarity:** 3
**Significance:** 3
**Originality:** 2
**Rating:** 4
**Confidence:** 4

**Summary:**

The authors consider the problem of robust estimation under the Heterogeneous corruption model, where the corruption across samples is independently but not identically distributed. They derive upper and lower bounds on the minimax rates (and minimax PAC rates) for several canonical estimation problem such as mean estimation for Gaussian distributions and linear regression in this heterogeneously robust corruption model.

**Questions:**

I would like to see more discussion on the O(\sqrtt{d}) gap in the regime where the dimension in not constant. Which of the bounds is conjectured to be tight by the authors and why?

**Ethical Concerns:**

["NO or VERY MINOR ethics concerns only"]

**Final Justification:**

I am maintaining my score. While I think the results are intersting, non-trivial, and should be published, the gap in high-dimensional settings prevents me from giving a higher score than my current one.

**Limitations:**

The authors adequately addressed the limitations and potential negative societal impact of their work.

**Paper Formatting Concerns:**

There are no major formatting issues in this paper.

**Quality:**

3

**Strengths And Weaknesses:**

Strengths: I find this particular model of heterogeneous robust corruption to be interesting and well supported by practical scenarios. The results are clearly presented and intuitions are given for the algorithms attaining the upper bounds and the lower bounds techniques. The difficulty of this scenario w.r.t. classical robust estimation is captured in the introduction, and the approaches of the authors are adjusted to retrieve non-trivial bounds.
Weaknesses: While I do think the results are intersting, they seem to be quite loose. I particularly refer to the results of Gaussian mean estimation and Linear Regression, which suffer from a O(\sqrt{d}) deficit between the upper and lower bounds, which is only tight in the non interesting constant dimension regime.

---

> ### Author Rebuttal · Authors · 2025-07-31
>
> We acknowledge that closing this gap in high dimensions is a very interesting open problem. However, we want to emphasize (as some reviewers also point out), that obtaining a $O(\sqrt{d})$ gap is already quite nontrivial, as all previously known techniques would achieve a $O(d)$ gap.
>
> From a technical perspective, there are new, interesting behaviors unique to these high dimensional settings that present as concrete barriers to reducing this gap, and it seems like neither bound is tight for all regimes. In the full version of the paper, we will expand upon the types of barriers we found, and why we believe that fundamentally new ideas are needed to obtain fully instance-optimal bounds. As a first step in this direction, in the full version of the paper, we will include a refined version of the lower bound technique which properly interpolates between Le Cam and Assoaud’s lemma, i.e., interpolates between $2$ to $2^d$ hypotheses. We believe that analyzing this technique properly gives us a promising avenue to obtaining tighter lower bounds for all regimes.

---

> > ### Comment · Reviewer_9oan · 2025-08-05
> >
> > Thank you for the response. The fact that neither bound is tight for all regimes is important and should be highlighted already in this version, even economically due to the lack of space.

---

### Official Review · Reviewer_1wmi · 2025-06-30

**Clarity:** 3
**Significance:** 3
**Originality:** 4
**Rating:** 5
**Confidence:** 3

**Summary:**

This paper introduces a novel framework for robust estimation under heterogeneous corruption, where each data point can be corrupted with a known, but potentially distinct, probability $\lambda_i$. This model is a generalization of the standard Huber contamination model, which assumes a uniform corruption rate.  The authors provide a theoretical analysis for several fundamental estimation problems within this framework. For bounded mean estimation, they establish tight minimax optimal rates. For multivariate Gaussian mean estimation and linear regression, they establish minimax rates up to a factor of $\sqrt{d}$, where $d$ is the dimension. The paper proposes two main types of estimators: a simple and intuitive thresholding-based method, and a more refined per-sample reweighting scheme. The results suggest that the optimal estimators may effectively discard samples whose corruption rates exceed a certain threshold, which is determined by the empirical distribution of the corruption rates themselves.

**Questions:**

1. The paper's main theoretical limitation is the $\sqrt{d}$ gap in the high-dimensional settings (Theorem 4 and 5). The authors suggest this might be an "artifact of the analysis".  Could you provide more intuition on why the current analysis techniques (specifically, the uniform convergence bounds for the weighted Tukey median in Appendix C.1) might be losing this factor? Do you have a conjecture on whether the upper or lower bound is closer to the true rate, and what new analytical tools might be needed to close this gap?

2. The framework assumes that the corruption rates $\lambda_i$ are known precisely. How would the proposed rates change if one only had access to estimates $\hat{\lambda}_i$ such that, for instance, $|\hat{\lambda}_i - \lambda_i| \leq \varepsilon$ for all $i$? Would the error from estimating $\lambda_i$ dominate the statistical error rates derived in the paper, or would it add a lower-order term? A brief formal discussion on this point would strengthen the paper's practical relevance.

3. The paper proposes both a simple thresholding estimator and a more complex reweighting estimator (Algorithm 1). For bounded mean estimation, the thresholding estimator is sufficient to achieve the minimax rate. The experiments in Figure 2a show only "marginal improvement" from the reweighting scheme.  Could the authors provide more insight into the scenarios where the additional complexity of the optimal reweighting scheme is justified? Is there a specific structure in the distribution of $\lambda_i$ values where reweighting provides a significant practical advantage over simply choosing an optimal threshold?

4. In the proof of the upper bound for the weighted Tukey median (Proposition 1, Appendix C.1), the result relies on Proposition 4, which provides a weighted generalization bound. The bound on the Rademacher complexity in Lemma 2 has a dependency on $\|w\|_2$.  This $\|w\|_2$ term eventually leads to the $\frac{d}{N(t)}$ term for the thresholding estimator. Is this dependence on $\|w\|_2$ fundamental, or is it possible that a different analysis technique could yield a bound that depends on the weights w in a more nuanced way, potentially helping to close the  $\sqrt{d}$ gap?

**Ethical Concerns:**

["NO or VERY MINOR ethics concerns only"]

**Final Justification:**

I am maintaining my score.

The paper introduces a novel and interesting framework for robust estimation with heterogeneous corruption. While the primary weakness is the $\sqrt{d}$ gap in high-dimensional settings, the authors convincingly argued that their result is a non-trivial improvement over prior work. Their plan to refine the lower bound is reasonable. The rebuttal successfully clarified all my concerns.

**Limitations:**

Yes.

**Paper Formatting Concerns:**

No.

**Quality:**

3

**Strengths And Weaknesses:**

### Strengths


1. **Originality and Significance:** The formulation of the robust estimation problem with heterogeneous corruption rates is novel and significant. It addresses a practical limitation of existing models that assume homogeneous corruption, making the framework relevant for applications in distributed settings like federated learning, crowdsourcing, and sensor networks.



2. **Theoretical Depth and Rigor:** The paper provides a comprehensive theoretical analysis, deriving minimax rates for several key problems. The proofs for the upper and lower bounds are rigorous. For the bounded mean estimation problem, the derived rate is shown to be tight, which is a strong result. The use of both Le Cam's and Assouad's methods for the lower bounds demonstrates a sophisticated handling of the technical challenges.


3. **Proposed Estimators:** The paper proposes both a simple, computationally efficient thresholding estimator and a more complex reweighting scheme (Algorithm 1). The thresholding approach is intuitive and shown to be minimax optimal (up to constants), providing a clear and actionable strategy. The introduction of weighted versions of Tukey depth and regression depth is a natural and elegant extension of classical robust methods to the heterogeneous setting.





### Weaknesses


1. **Gap in Bounds for High Dimensions:** The most significant limitation is the  $\sqrt{d}$ gap between the upper and lower bounds for multivariate Gaussian mean estimation and linear regression.  While the authors are transparent about this gap, it means the characterization of the minimax rate in high dimensions is incomplete. It remains unclear whether this is an artifact of the analysis or a true phenomenon.


2. **Assumption of Known Corruption Rates:** The entire framework relies on the assumption that the corruption rates $\lambda_i$ are known exactly.  The authors mention that their bounds generalize to approximate knowledge of $\lambda_i$, but this is not formally detailed. In many practical scenarios, these rates may need to be estimated, which would introduce an additional layer of error not accounted for in the current analysis.


3. **Limited Practical Demonstration:** The experimental evaluation is presented as "preliminary". While the results in Figure 2 do support the theoretical claims by showing improvement over baselines, they are limited to low-dimensional synthetic data. Furthermore, the practical advantage of the more complex reweighting scheme over the simple thresholding method is described as "marginal" for the bounded case, which might underwhelm practitioners.

---

> ### Author Rebuttal · Authors · 2025-07-31
>
> We thank the reviewer for their review.
>
> 1) Please see the response we posted to reviewer 9oan.
>
> 2) See response to Reviewer hemm. In general, our methods are able to easily tolerate multiplicative errors in knowing the $\lambda_i$. We did not consider the exact form of additive error that the reviewer proposes, but in some sense it seems less natural to have additive error when $\lambda_i$ is small. However, we believe exploring this could be an interesting direction.
>
> 3) This is indeed an interesting question. In theory, the reweighting scheme subsumes the thresholding estimator. While we use the thresholding estimator for proving the minimax bounds, we present the more complex estimator to demonstrate the possible constant-factor advantages in the bounded setting. In fact, we conjecture that in the Gaussian high-dimensional setting, the weighted Tukey estimator should outperform the thresholding estimator, at least in some regimes.
>
> 4) While the weighted Tukey median likely does sometimes outperform thresholding-based estimators in the high dimensional setting, there is a fundamental difficulty with obtaining tight bounds in all regimes with any weighted Tukey median-based technique. In particular, in the semi-verified regime (as defined on lines 65-74), one can show that no weighted Tukey median will be able to obtain optimal rates in all parameter regimes. Rather, it seems that a “two-step” process will be necessary to obtain tight statistical rates. This is one of the concrete barriers we encountered that we alluded to in the response to Reviewer 9oan, and in the next version of the paper, we will make this connection more explicit.

---

> > ### Comment · Reviewer_1wmi · 2025-08-04
> >
> > Thank you for the detailed rebuttal. Your responses have satisfactorily addressed my questions, particularly regarding the $\sqrt{d}$ gap and the role of the different estimators. Please include the planned revisions in the paper. My positive assessment of the paper remains.

---

### Official Review · Reviewer_VCX4 · 2025-07-03

**Clarity:** 2
**Significance:** 3
**Originality:** 3
**Rating:** 4
**Confidence:** 3

**Summary:**

The paper proposes and studies the model of heterogeneous corruption where the $i$-th sample in the dataset has a different (assumed to be known) corruption level $\lambda_i$ and the $i$-th point is generated by (i) determining with probability $\lambda_i$ whether the sample is inlier and (ii) in case of inlier the sample is drawn from the inlier distribution, otherwise it is drawn from a different distribution (given the previous samples).

The paper establishes the minimax rates for several estimation problems under this model: Mean estimation for multivariate bounded distributions, mean estimation for multivariate Gaussians with identity covariance and multivariate linear regression. For the last two there is a gap of $\sqrt{d}$ between the upper and lower bounds.

The upper bounds all involve the function $f(\lambda,k)=\min_{t \in [0,1]}(k/\{\lambda_i \leq t\}+t^2)$ which is related to the best effective corruption rate. The upper bounds can be achieved by a simple algorithm that chooses a corruption level $t$ and runs an estimator on the subset of the data data corresponding to $\lambda_i \leq t$ (where $t$ is chosen to minimize the error). The authors also show alternative methods based on reweighting the points (for the bounded distributions case they show a method that optimizes the weights of the points based on explicit calculations, and for the other problems they show variants of the Tukey median and Tukey regression depth). The lower bounds are built on Le Cam's two point method.

**Questions:**

The third bullet in the definition of the contamination model is a bit stronger than Huber's model with heterogeneous corruption levels, since outliers are drawn adaptively. Is this adaptivity necessary to assume for the lower bounds?

**Ethical Concerns:**

["NO or VERY MINOR ethics concerns only"]

**Final Justification:**

The authors have provided clear explanations to my points. After reviewing the other rebuttals and discussions my view of the paper remains positive, I thus keep my original score.

**Limitations:**

yes

**Paper Formatting Concerns:**

No issues

**Quality:**

3

**Strengths And Weaknesses:**

Strengths:

- Estimation with heterogeneous data is well motivated. The paper gives the first non-trivial results for basic estimation problems that remained unstudied.
- The paper provides experiments showing the advantage of the proposed algorithms.

Weaknesses:

- The bounds on the minimax squared $\ell_2$ error are tight only for the bounded support distributions and univariate Gaussians and has a $\sqrt{d}$ gap otherwise. This is still better than naive coordinate-wise methods but means that the precise bound is still unknown.
- Some of the narrative about the upper bound method was somewhat unclear to me even after the explanation of page 2. If the simple algorithm of running the homogeneous contamination estimator for the best level achieves the claimed error upper bound is there a theoretical advantage of the other methods presented or they are shown as alternative methods?

---

> ### Author Rebuttal · Authors · 2025-07-31
>
> We thank the reviewer for their review and apologise for the confusion regarding the discussion of weighted estimator in the text. We clarify that indeed the thresholding method, a strict subset of the weighted estimator, is used to obtain the bounds provided in this work. However, we include the weighted estimator as we believe that it might be important in establishing a tighter bound in the high-dimensional Gaussian setting in some regimes. We shall change the current exposition to reflect this point more clearly.
>
> The reviewer also asks about the adaptivity in the contamination model. We emphasize that our lower bounds do not require adaptivity, whereas our upper bounds hold even against adaptive adversaries. We consider the adaptive adversary in large part only because such adversaries are standard in the robust statistics literature [21]. However, it is an interesting question to fully characterize the power of adaptivity in such heterogeneous settings, as has been done in the standard homogeneous setting.
>
> Regarding the $O(\sqrt{d})$ gap, please see the response we posted to Reviewer 9oan.

---

> > ### Comment · Reviewer_VCX4 · 2025-08-05
> >
> > Thank you for the response. It addresses my points. I do not have further questions for the authors.

---

### Official Review · Reviewer_hemm · 2025-07-04

**Clarity:** 2
**Significance:** 3
**Originality:** 3
**Rating:** 5
**Confidence:** 3

**Summary:**

The paper presents a generalization of the Huber contamination model to heterogenous known per-sample corruption rates and presents estimators and proofs of optimality (up to a constant factor) for mean estimation in two settings (bounded distributions and Gaussians) as well as linear regression with Gaussian covariates.

**Questions:**

How frequently is knowing $\lambda$ (or an accurate approximation) realistic in settings such as federated learning?

**Ethical Concerns:**

["NO or VERY MINOR ethics concerns only"]

**Final Justification:**

I maintain my score of accept given the paper's significant theoretical contributions.

**Limitations:**

yes

**Quality:**

3

**Strengths And Weaknesses:**

### Strengths

Robust mean estimation is an important problem; the author's heterogenous contamination model is applicable to practical settings such as federated learning. The optimal estimators presented are valuable contributions; the proofs of optimality presented are clear and helpful.

### Weaknesses

As noted by the authors, the estimators presented are only tractable on low-dimensional data, and require that corruption rates for each sample are known, which may not always be realistic in practice.

The paper has a large number of minor typos and grammatical errors:
- "are not independent can dependent" on line 197
- $||\mu_{P_1} - \mu_{P_2}||^2_2$ should be $||\mu_{P_0} - \mu_{P_1}||^2_2$ on line 215
- "This conditions" should be "This condition" on line 228
- "get" -> "get a" on line 230
- $\hat{\mu}_S(\boldsymbol{Z}, w(t))$ is generally used instead of the $\hat{\mu}_S(\boldsymbol{Z}, t)$ from Equation 20 that appears to be intended

---

> ### Author Rebuttal · Authors · 2025-07-31
>
> We appreciate the reviewer for pointing out this practical question. It can indeed be challenging to know the corruption rates in practice. It may be noted that we just need a constant factor approximation $\hat \lambda_i$ of the corruption rate for our bounds to hold (i.e., $\lambda_i \leq c \hat \lambda_i$). For this reason, it is common in the robust estimation literature to assume that we roughly know the corruption rate. In practice, corruption rates can often be estimated from past performance. For example, the reviewer mentions the setting of federated learning. In many standard federated learning setups, such as those which arise in learning over an IoT sensor network, one can monitor the response of nodes to estimate individual corruption rates. In addition, our assumption of $\lambda$ being known is still a useful modeling choice. The theory we develop allows us to understand how the estimation error degrades as a function of the corruption rates; this can help answer practical questions such as whether it is better to replace some bad sensors, or to focus on expanding the network with possibly lower quality sensors.

---

### Note · Authors · 2025-08-16

We thank the reviewers for their astute comments. We are greatly encouraged by the reviews and we hope that the reliable machine learning and statistics community in general receives our work with similar enthusiasm. We believe that real world systems naturally exhibit heterogeneity in reliability across different data sources and leveraging this heterogeneity can enhance the performance of these systems. In terms of theory, we have extended classical statistical techniques for dealing with heterogeneity in terms of minimax bounds. There are several avenues for future research and we believe presenting our work at NeurIPS would allow us to communicate these avenues to other researchers as well as motivate them to work on this interesting topic.

---

### Decision · Program_Chairs · 2025-09-17

**Decision:**

Accept (poster)

**Comment:**

This paper is a contribution in robust statistics, studying a generalization of the Huber contamination model where the corruption rate depends on the (index of) the sample and is known a-priori. The paper studies mean estimation (bounded, Gaussian) and linear regression (Gaussian), and show either up-to-constant tight minimax rate results, or new results that have a $\sqrt{d}$ factor gap.

Reviewers agree that the model and problem setting are both of interest to the NeurIPS community. While there is still a $\sqrt{d}$ gap for some problems, this is already an improvement as prior techniques would yield a $O(d)$ gap.

I recommend acceptance of this paper as there are no soundness concerns and the results are of interest to the community.